

# Understanding the Effect of Revegetated Shrubs on Energy, Water and Carbon Fluxes in a Desert Steppe Ecosystem Using STEMMUS-SCOPE Model

Enting Tang[1], Yijian Zeng[1], Yunfei Wang[1,4], Zengjing Song[1,5], Danyang Yu[1], Hongyue Wu[2,3], Chenglong
Qiao[2,3], Christiaan van der Tol[1], Lingtong Du[2,3], Zhongbo (Bob) Su[1]

[1]Faculty of Geo-Information Science and Earth Observation (ITC), University of Twente, 7500 AE Enschede, the Netherlands
[2]Breeding Base for State Key Laboratory of Land Degradation and Ecological Restoration in Northwest China, Ningxia University, Yinchuan, China
[3]Key Laboratory for Restoration and Reconstruction of Degraded Ecosystem in Northwest China of Ministry of Education,
Ningxia University, Yinchuan 750021, China
[4]School of Hydraulic and Civil Engineering, Zhengzhou University, Zhengzhou, 450001, China
[5]Chongqing Jinfo Mountain Karst Ecosystem National Observation and Research Station, School of Geographical Sciences, Southwest University, Chongqing 400715, China

*Correspondence to*: Zhongbo Su (z.su@utwente.nl), Yijian Zeng (y.zeng@utwente.nl), Lingtong Du (dult80@nxu.edu.cn)

**Abstract.** Revegetation is one of the most effective ways to combat desertification and soil erosion in semiarid and arid regions. However, the perturbation of revegetation on ecohydrological processes, particularly its effects on the interplay between hydrological processes and vegetation growth under water stress, requires further investigation. This study evaluated the effects of revegetation on the energy, water and carbon fluxes in a desert steppe in Yanchi County, Ningxia Province, Northwest China, by simulating two vegetated scenarios (shrubs-grassland ecosystem and grassland ecosystem) using STEMMUS-
SCOPE model. The model was validated by field observations from May to September of 2016-2019. The evaluation of revegetation effects relied on comparing simulated fluxes between two vegetated scenarios in 2016 and 2019. Higher leaf area index and root water uptake of C3 shrubs (*Caragana Intermedia*) resulted in increased carbon fixation (+ 82 %) and transpiration (+ 99 %) of the shrubs-grassland ecosystem compared to C3 grassland ecosystem. In both scenarios, turbulent energy was dominated by latent heat flux, which was stronger in the shrubs-grassland ecosystem (+ 13 %). With the remarkable
increase in transpiration, revegetation induced soil water losses, especially the soil water content within the 0-200 cm soil depth (− 19 %), and intensified the excess of water consumption over the received precipitation. These results emphasize the importance of accounting for energy and water budget in water-limited ecosystems during ecological restoration, to prevent soil water depletion. As an example, the consequence of increased transpiration should be further examined.

## 1 Introduction

Global efforts in revegetation have been made to combat climate change and desertification. For example, satellite data reveals that the revegetation programs in China have contributed about 10.5 % of the increased global greening during 2000-2017 (Chen et al., 2019). This large-scale revegetation program ('Grain-to-Green') was initiated to improve the ecosystem service of degraded desert steppe in northern China since the 1990s (Liu et al., 2021). On the one hand, it is proven effective in controlling soil erosion and enhancing carbon sequestration (Liu et al., 2021; Zhang et al., 2018). On the other hand, the





conflict between the water deficit and the development of the shrub community has become an increasing concern, particularly in arid and semiarid lands where ecosystems are fragile and suffering intense water stress (D'Odorico et al., 2012; Tian et al., 2017; Huxman et al., 2005; Zhang et al., 2018).

      For example, the revegetation in China's Loess Plateau has increased the net primary productivity and evapotranspiration, but
the ecosystem is approaching sustainable water resource limits (Feng et al., 2016). Specifically, field studies reported that the revegetation leads to the depletion of soil moisture (Liu and Shao, 2015; Jia et al., 2017), formation of the dry soil layer (Fu et al., 2012; Jia et al., 2017; Jian et al., 2015; Gao et al., 2023) and reallocation of the energy partitioning along with changes in vegetation distribution and canopy structure (Chen et al., 2015). Therefore, implementing revegetation programs but ignoring their long-term effects on energy, water and carbon balance may act contrary to ecologically sustainable development.
Quantitative assessment of energy, water, and carbon fluxes is essential for evaluating the impact of revegetation, including the determination of water resource limits and optimal plant coverage for revegetation (Fu et al., 2012; Feng et al., 2016). However, the lack of long-term observations makes it difficult to reproduce the energy, water and carbon cycles of the ecosystems before and after the revegetation practice. To overcome this challenge, process-based land surface models (LSMs) can provide a better understanding of the energy-water-carbon flows of ecosystems (Du et al., 2021; Gong et al., 2016).


      The past few decades have seen the rapid development of LSMs for dryland ecosystems based on the soil-vegetation-atmosphere transfer continuum (SVAT) (Tague et al., 2004; Ivanov et al., 2008; Fatichi et al., 2016; Niu et al., 2020). It is widely believed that the dominant constraint of vegetation development in the semiarid region is soil water availability, which manifests itself in regulating photosynthesis, evapotranspiration and root distribution (Camargo and Kemanian, 2016; Fan et
al., 2017). In this context, accurate soil water modelling in LSMs is critical for the overall model performance in predicting energy, water and carbon fluxes. However, some existing deficiencies in the soil water modelling include: (a) computing the soil water content with a simple "bucket" approximation (e.g., RHESSys and Biome-BGC), (b) defining maximum root water uptake capacity with empirical constants (e.g., CLM and tRIBS + VEGGIE) or by using a direct function based on soil water availability (Zeng et al., 1998; Tague et al., 2004; Zhang et al., 2013; Fisher et al., 2014; Newman et al., 2006). The "bucket"
model may overlook the soil water movement through different layers due to its simple representation of the vertical soil profile and root structure (Romano et al., 2011; Du et al., 2021). Therefore, the physics-based model that considers the simultaneous transfer of water, vapor, and heat as well as root water uptake across the soil profile (with multiple layers) is preferable in the vegetated semiarid region (Wang et al., 2021b; Zeng et al., 2011a; Yu et al., 2020, 2016).

The coupled STEMMUS-SCOPE model (STEMMUS - Simultaneous Transfer of Energy, Momentum, and Mass in Unsaturated Soil; SCOPE - Soil-Canopy Observation of Photosynthesis and Energy) (Wang et al., 2021b) can simulate the profile of dynamic root length density for estimating root water uptake and the hydraulic resistance from soil, to root, stem, and to leaf. Thanks to the inclusion of plant hydraulic connection between soil and leaf, the coupled model realizes the

influence of soil moisture variation on photosynthetic and stomatal behaviours, which facilitates the investigation of water

stress effects on vegetation functioning. The coupled model also considers the compensatory root water uptake, by which uptake from sparsely rooted but well-watered parts of the root zone compensates for stress in other parts.

The objective of this study is to quantify the effects of planting *Caragana Intermedia* on the energy, water, and carbon fluxes in a semiarid desert steppe ecosystem in northwestern China with STEMMUS–SCOPE model. The approaches to achieve this objective are as follows. First, the contributions and leaf area index of shrubs and grasses were defined (Section 2.3.1 and

2.3.2) and they were further used to construct two vegetated scenarios (shrubs-grassland ecosystem and grassland ecosystem) (Section 2.3.3). Secondly, the sensitivity of seven critical parameters to fluxes was analysed using Morris method (Section 2.4.1 and Section 3.1). Thirdly, STEMMUS-SCOPE was calibrated and validated with observations over May-July in 2018 and May-September in 2016, 2017 and 2019, respectively (Section 3.2). Finally, the fluxes simulation results from two scenarios were compared and their differences were analyzed (Section 3.3).

## 80    2 Materials and Methods

### 2.1 Study site

The study area (107°29′37″ E, 37°49′46″ N) is located in Yangzhaizi village of Yanchi County, Ningxia Province, which is a typical agricultural-pastoral ecotone with a mid-temperate semiarid continental climate (Fig. 1). The mean annual air temperature (1958-2017) was 8.34 °C. The mean annual precipitation is 296.99 mm, and about 80 % of the rain falls between

June and September (Jia et al., 2018). Since the 1990s, the grazing-prohibiting policy and the revegetation program have been implemented in the study area to combat desertification.

The study site is a 30 m × 30 m fenced plot characterized by revegetated shrubs and natural grassland. The shrub strips were planted at an interval of 6-7 m, and the average distance between two neighbouring shrubs in each strip was less than 1 m (Fig.

S2). Between and below the canopy of shrubs, sparse grasses grow and soil moisture sensor (SM150, DELTA-T, UK), soil temperature sensor (107-L, BetaTherm, DE) and heat flux plate (HFP01, Hukseflux, NL) were installed at 10 cm soil depth under the grasses. The EC flux tower has an open path $CO_2/H_2O$ analyzer (LI-7500, LI- COR Inc., USA) and a 3D ultrasonic anemometer (Wind Master Pro, Gill, UK), a tipping bucket rain gauge (TE525 MM-L, Texas Electronics, USA), net radiometer (CNR-4, Kippen&zonen, NL), temperature and relative humidity probe (HMP45C, CSL USA), and is surrounded by *Caragana*

*intermedia* perennial shrubs and native herbaceous plants (Fig. 1). The active growing season of the shrubs and grasses lasts from May to September. The predominant soil texture is characterized as aeolian sandy soil.



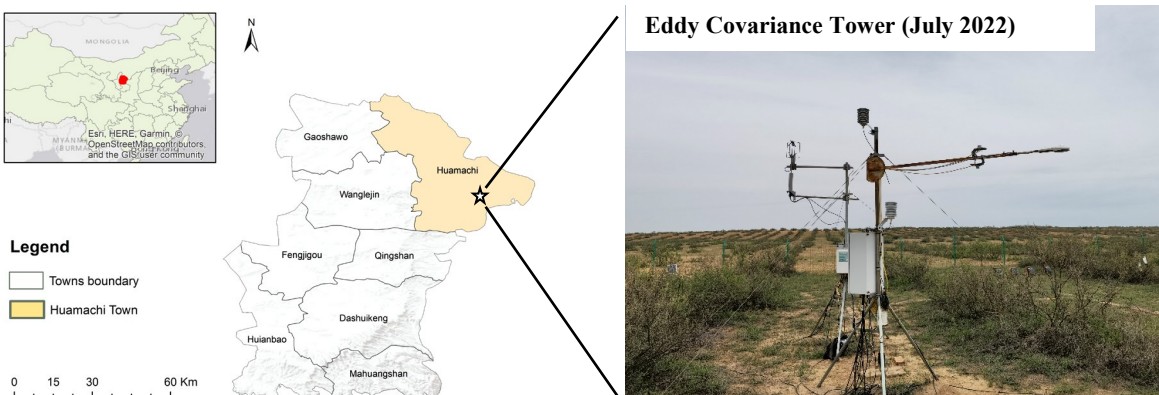

**Figure 1. Overview of the study area. The red area in the map (from OpenStreetMap) is Yanchi County located in China. The white star denotes the field station located in Yangzhaizi village in Yanchi County, Ningxia Province.**


## 2.2 STEMMUS-SCOPE model

Only the major modules in STEMMUS-SCOPE used in this study are described in Eqs (S1) - (S26). The original descriptions of model components and other applications are elaborated by Bayat et al. (2019), Van der Tol et al. (2009), Wang et al. (2021b), Yang et al. (2020), Zeng et al. (2011a, 2011b). The STEMMUS-SCOPE is employed to simulate the energy, water

and carbon fluxes for the two ecosystems, representing the scenarios before and after revegetation. The following assumptions are made for modelling: (i) A land unit is structured as a vertical continuum, which consists of shrub or grass and soil column; (ii) As the groundwater is more than 6 m below the surface in the study area (Du et al., 2021), the boundary condition at the bottom of the soil column (i.e., 5 m depth) is set as gravity drainage; (iii) the soil texture is vertically homogeneous. In this context, the model runs at a half-hourly time step, with the input of meteorological forcings, soil hydraulic and plant trait

parameters (Table S1). Model outputs are compared to data from the EC tower, which includes net radiation, latent heat flux, sensible heat flux, and gross primary productivity. Additionally, ground measurements of soil moisture, soil temperature, and corrected surface soil heat flux (using Eq. S9) are used for validation.

## 2.3 Simulation scenarios design

### 2.3.1 Land cover classification

Two vegetation covers both contribute to the EC observations and must be reflected in the model parametrization. Therefore, a classification map is needed to derive the fractional vegetation cover for a more accurate modelling. The Supervised Classification Method in ERDAS 2020 was used to determine the fractional cover of shrubs, grasses and bare soil based on an image taken by unmanned aerial vehicle (Fig. S2).






STEMMUS-SCOPE considers the soil-root-canopy continuum, and quantifies the amount of energy received and water evaporated from its canopy and soil based on the leaf area index and leaf inclination (e.g., with four temperature variables: sunlit/shaded leaf temperatures, sunlit/shaded soil surface temperatures). Here, we assume that the 40 % coverage of bare soil distributed in each simulated soil-root-canopy continuum (i.e., either shrub or grass) and make the approximated contribution

of 58.33 % for shrubland and 41.67 % for grassland (Table 1).

**Table 1. Fractional coverage of shrubs, grasses and bare soil and the approximated contributions from shrubs and grasses.**

| Land Cover | Number of pixels | Fractional coverage in field | Contribution in simulated fluxes* |
|---|---|---|---|
| Shrub | 268,325,3 | 35 % | 58.33 % |
| Grassland | 195,308,4 | 25 % | 41.67 % |
| Bare Soil | 317,921,8 | 40 % | Implicitly included for either Shrub grid or Grass grid |
| Instrument | 354,78 | / | / |

*This contribution will be further used to aggregate the simulated fluxes of the shrubs-grassland scenario (including contribution of bare soil evaporation).

**2.3.2 Reconstructed LAI**

Leaf area index (LAI) is a critical variable in calculating the Gross Primary Productivity (GPP) and latent heat flux (LE) in the STEMMUS-SCOPE. The MODIS 4-day LAI data ($LAI_{MODIS}$) during 2016-2019 was extracted from the Google Earth Engine Platform. Further, we applied the Harmonic Analysis of Time Series (HANTS) algorithm in MATLAB to smooth the outliers (Supplement Section 2).


The study site is a 30 m × 30 m plot with two species, while $LAI_{MODIS}$ can only provide an overall LAI with 500 m spatial resolution. To achieve the simulated flux partitioning from two land covers, we simulated shrub grid and grass grid with the LAI of shrub ($LAI_{shrub}$) and LAI of grassland ($LAI_{grass}$), respectively (Fig. 3c). With only two field measurements in 2022, $LAI_{shrub}$ was corrected by multiplying smoothed $LAI_{MODIS}$ by 2.33 (Table S4). $LAI_{grass}$ was estimated by assuming it was $\frac{1}{4}$

of that of the shrubs, and by disaggregation of $LAI_{MODIS}$ with the followings:

    i.    $f_{shrub} * LAI_{shrub}(i) + f_{grass} * LAI_{grass}(i) + f_{baresoil} * LAI_{baresoil} = LAI_{MODIS}(i)$

    ii.    $f_{shrub} + f_{grass} + f_{baresoil} = 1$

    iii.    $LAI_{baresoil} = 0$

    iv.    $LAI_{shrub}(i) \approx 4\ LAI_{grass}(i)$ (Dan et al., 2020)

v.    $LAI_{grass}(i)$ should follow the temporal pattern of $LAI_{MODIS}(i)$ and it was ~0.5 m² m⁻² (Yang et al., 2019; Dan, 2020).





where $f_{shrub}$, $f_{grass}$ and $f_{baresoil}$ are the fractional cover of shrubs, grasses and bare soil, respectively. With the above constraints, $LAI_{grass}$ shown on Fig. 2 was generated by the HANTS algorithm (Table S4 and Fig. S3).

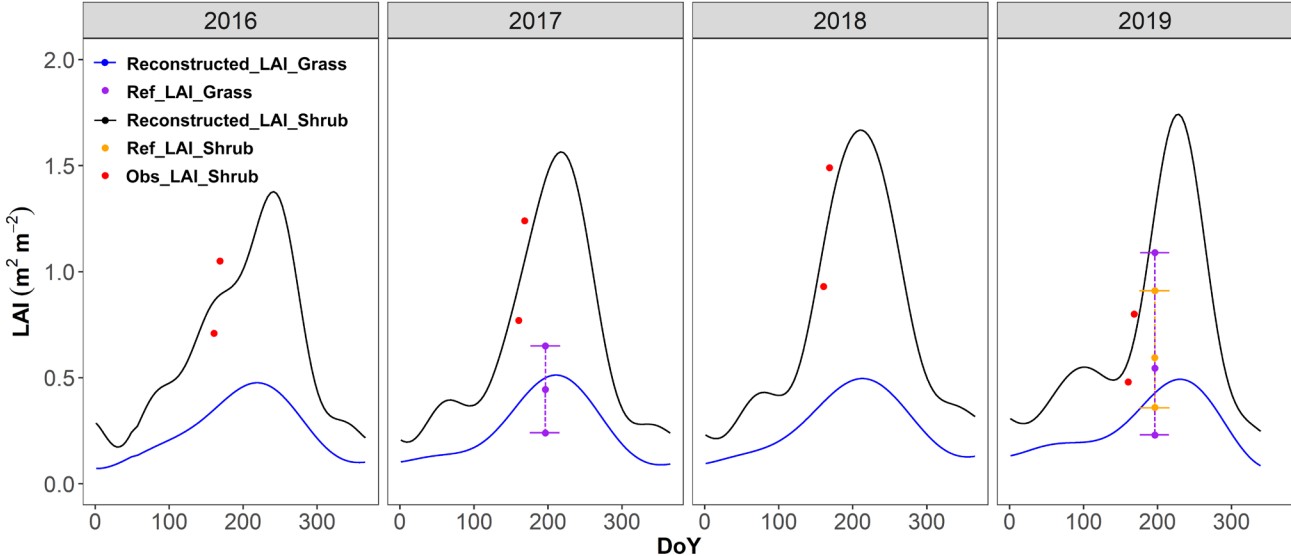

**Figure 2. Reconstructed LAI of shrubland and grassland from 2016 to 2019. The red dots (Obs_LAI_Shrub) are the actual LAI of shrubs that were calculated from bias correction based on the ratio derived in 2022 (Table S4). The purple dots and dotted lines (Ref_LAI_Grass) represent the ranges of measured LAI of the nearby grasslands (Yang et al., 2019; Dan, 2020). The yellow dots and dotted lines (Ref_LAI_Shrub) represent the ranges of measured LAI of the nearby shrublands (Dan, 2020).**

### 2.3.3 Scenarios design

Two sets of canopy parametrization schemes (Table S1) together with the reconstructed LAI were used to simulate the fluxes of shrub grid and grass grid, respectively (Fig. 3c). We assumed the shrubland and grassland had the same meteorological environment; likewise, the same initial conditions of soil temperature and soil water content were assigned to the two land covers (Table S2). To accurately depict the fluxes of mixed surfaces containing both shrubs and grasses, the simulated fluxes from the shrubland and grassland simulations were partitioned based on their respective contributions, as demonstrated in Fig. 3(b). For instance, the total evaporation, transpiration and gross primary productivity (GPP) were the sum weighted by their contributions. As an example:

$$GPP = C_{shrub} \, GPP_{shrub} + (1 - C_{shrub}) \, GPP_{grass} \tag{1}$$

The same partitioning method as GPP was applied to net radiation, latent heat flux and sensible heat flux, in which the latent heat was converted into evaporation flux using the evaporation heat at 20 °C.



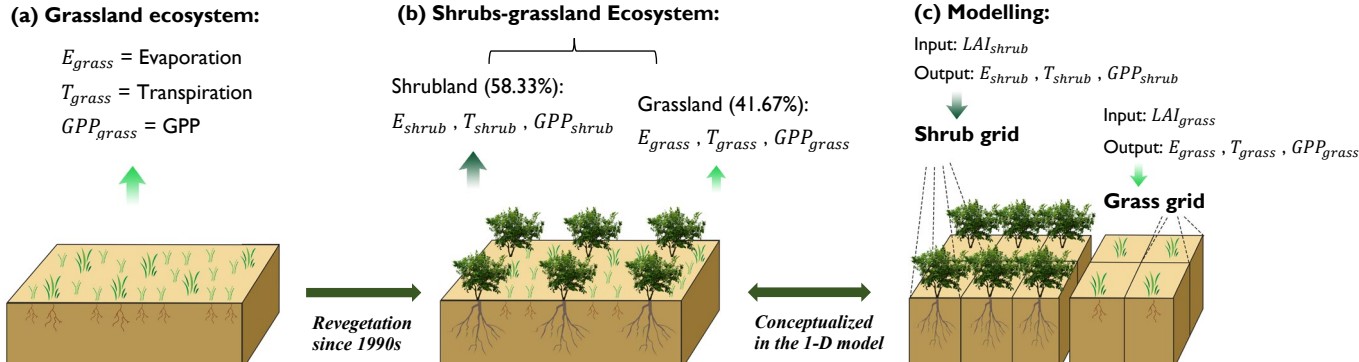

**Figure 3. Conceptual diagram of (a) native grassland scenario before revegetation (hereafter is denoted as grassland ecosystem), (b) realistic scenario with mixed shrubs and grasses (hereafter is denoted as shrubs-grassland ecosystem) and (c) conceptualization in the model. The bare soil evaporation is implicitly included in $E_{shrub}$ and $E_{grass}$, where $E$ is the evaporation.**

## 2.4 Model calibration and validation

### 2.4.1 Sensitivity analysis

Morris global sensitivity analysis (SA) was applied to evaluate the sensitivity of model simulations to the variation of input parameters. Morris SA, a screening-based method, can give a qualitative ranking of parameters at a relatively low computational cost (Herman et al., 2013). The most influential parameters for STEMMUS model and SCOPE model identified by other literature are: Maximum carboxylation rate $(V_{cmax})$, Ball-Berry stomatal conductance parameter $(m)$, Leaf inclination (LIDFa), Residual soil water content $(\theta_r)$, Van Genuchten parameters ($n$ and $\alpha$), Saturated hydraulic conductivity $(K_{sat})$ (Wang et al., 2021a; Verrelst et al., 2015). Note that SA was only done for parameters of the shrubland simulation and the range of soil parameters was set for the specific sandy soil (Table 2). The energy and carbon fluxes from each run were the composited fluxes aggregated from the shrubland trajectories and grassland simulations with fixed parameters listed in Table S1.

**Table 2. Range of critical parameters for SA.**

| Module | Parameters | Description | Units | Range |
|--------|-----------|-------------|-------|-------|
| Canopy | $V_{cmax}$ | Maximum carboxylation rate of C3 Shrub | μmol m⁻² s⁻¹ | [60[a], 250] |
| | $m$ | Default values of Ball-Berry slope | - | [2, 20] |
| | LIDFa | Default values of leaf inclination | - | [-1, 1] |
| Soil | $\theta_r$ | Soil parameters of sandy soil in similar sites (e.g., with similar species and/or soil texture) in Northwest China | m³ m⁻³ | [0.004[b], 0.035[c]] |
| | $n$ | | - | [1.38[b], 2.09[b]] |
| | $\alpha$ | | m⁻¹ | [0.0028[b], 0.03[d]] |
| | $K_{sat}$ | | cm d⁻¹ | [100, 300[b]] |

[a](Wang et al., 2017); [b](Gong et al., 2016); [c](Montzka et al., 2017); [d](Wei et al., 2019)






As the start of Morris SA, we had a $n$-dimension $p$-level orthogonal input space and STEMMUS-SCOPE model $Y = y(x_1, x_2, ….. x_n)$. Parameters are assumed to be uniformly distributed in [0,1] and randomly take values from $\{0, 1/(p-1), 2/(p-1), … , 1\}$. A trajectory $Y(x_1,…, x_k)$ is then generated. The elementary effect ($EE$) of the i$^{th}$ input is calculated as:

$$EE_i = \frac{Y(x_1,…,x_{i-1},x_i+\Delta_i,x_{i+1},…,x_k)-Y(x_1,…,x_k)}{\Delta_i}$$ (2)

where $k$ is the number of parameters ($k = 7$) and $p$ is the number of levels ($p = 16$). $\Delta$ is the variation in the parameter $x_i$, predetermined as the multiple of $1/(p-1)$. Each input parameter in a trajectory is assumed to vary across $\Delta$, introducing $(k + 1)$ elementary effects. Only one input parameter was perturbed between two successive runs of the model (Fig. S4). To achieve the stability of the SA results, $r$ different trajectories ($r = 20$) were randomly sampled from the $p^k$ ($16^7$) sampling space. Thus, the total runs of the model are $r(k + 1)$. At the SA stage, the model ran 160 times and generated 160 $EE_i$ for the

simulations during May-July in 2018. The Morris analysis was achieved using the SAlib package in Python (Herman and Usher, 2017). Besides, the parameterization of the best-fit trail with the minimal normalized root mean square errors $RMSEn = \frac{RMSE_{SWC}}{\overline{Obs}_{SWC}} + \frac{RMSE_{LE}}{\overline{Obs}_{LE}} + \frac{RMSE_{GPP}}{\overline{Obs}_{GPP}}$ was identified (Groenendijk et al., 2011). The $\overline{Obs}_{SWC/LE/GPP}$ is the average values of observed SWC, LE and GPP throughout the investigation period, respectively.

**2.4.2 Performance metrics**

Root mean square error (RMSE) and coefficient of determination ($R^2$) were used to evaluate the quality of the model predictions.

$$RMSE = \sqrt{\frac{\sum_{i=1}^{n}(x_o - x_s)^2}{n}}$$ (3)

$$R^2 = 1 - \frac{\sum(x_o - x_s)^2}{\sum(x_o - \overline{x_o})^2}$$ (4)

where $x_s$ is the simulated value, $x_o$ is the corresponding observed value and $\overline{x_o}$ is the mean of observed values, $n$ is the number of data records.

**3 Results**

**3.1 Model sensitivity**

The soil hydraulic parameters represented the strongest main effect and interaction effect on the simulated fluxes (Fig. S5).

Specifically, we highlighted the most influential parameters for each flux: (i) LIDFa for net radiation (Rn), (ii) $\alpha$ and $n$ for ground heat flux (G), (iii) $m$ and $\alpha$ for latent heat flux (LE), (iv) $m$ for sensible heat flux (H), (v)



$V_{cmax}$ for GPP, (vi) α and $n$ for soil water content (SWC). Notably, it is observed that the simulated G is highly dependent on the VG-coefficients (i.e., α and $n$). A similar influential ranking of parameters: $m$ > VG-coefficients > $\theta_r$ > $K_{sat}$ > $V_{cmax}$ for LE and H simulations is observed.


The parameterization that yielded the minimal $RMSEn$ for shrub (grass) is as follows: $V_{cmax}$ = 199.33 (120), $m$ = 10.4 (7), LIDFa = − 0.47 (− 0.47), α = 0.0155, $n$ = 1.71, $\theta_r$ = 0.01 m³ m⁻³, $K_{sat}$ = 153.33 cm d⁻¹. However, in light of the reference values from studies in Yanchi County or studies involving similar species, $V_{cmax}$ = 120, $\theta_r$ = 0.006 m³ m⁻³, and $K_{sat}$ = 288 cm d⁻¹ were used in conjunction with the other parameters derived from the optimal parameterization for model calibration

and validation.

## 3.2 Model performance

Data collected from May to July 2018 was used to calibrate the model (Fig. S6-S8). Compared with the simulations considering only one land cover, the model performance was better in capturing the dynamic and magnitude of energy and carbon fluxes

when considering the mixed land covers (Fig. S8).

Data collected during May-September in 2016 and 2019, May-July in 2017 was used to validate the model simulations. The simulated energy fluxes showed satisfactory agreement with the observed values, with the $R^2$ all above 0.66 and with the RMSE ranging from 19.17 to 66.92 W m⁻² (Fig. 4a-4d). The simulated turbulent flux (i.e., LE+H) followed well the trend of

the measured values ($R^2$ = 0.86, RMSE = 60.22 W m⁻²).

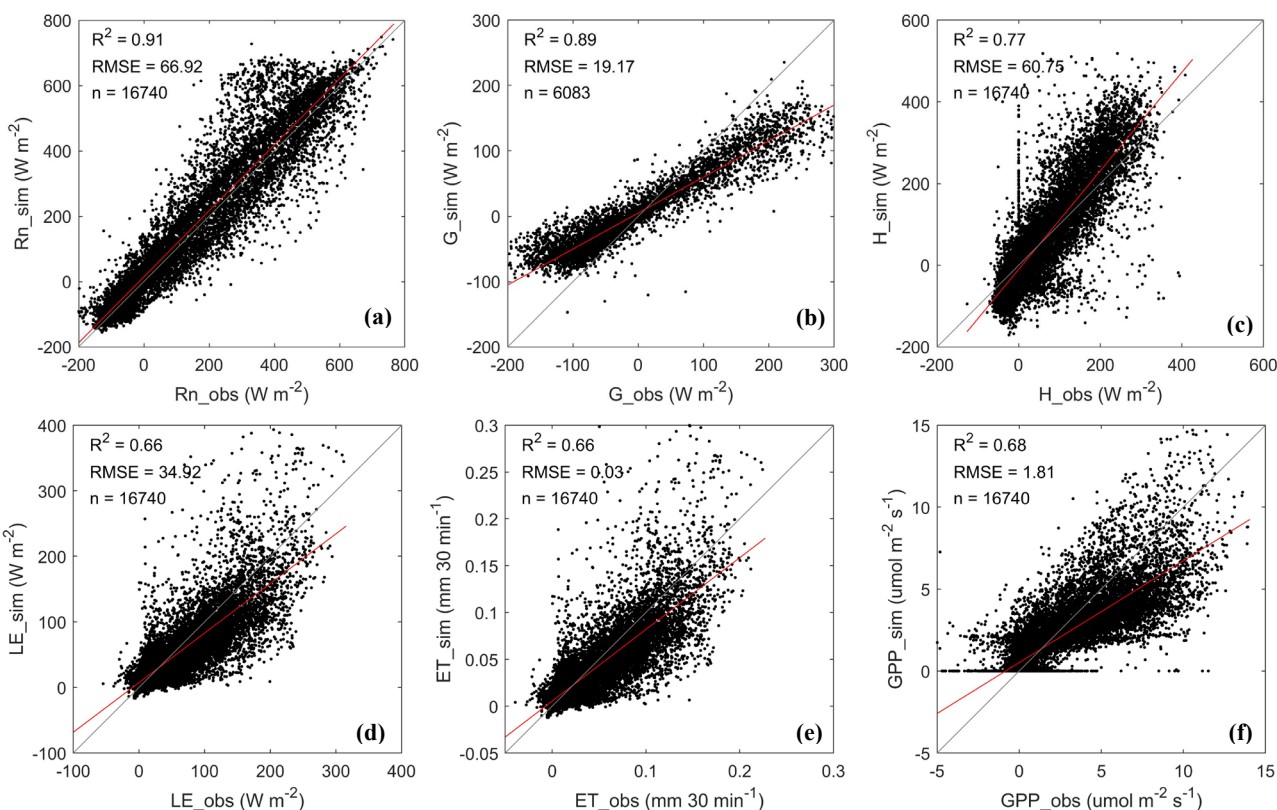

**Figure 4. Comparing the observed (obs) and simulated (sim) half-hour values of (a) Net radiation (Rn) , (b) Ground heat flux (G), (c) Sensible heat flux (H), (d) Latent heat flux (LE), (e) Evapotranspiration (ET) and (f) Gross primary productivity (GPP). The performance statistics are summarized in Table S5.**


The simulated soil water content (SWC) and soil temperature (Ts) were validated by the observed data from the grassland ecosystem (Fig. 5) as the sensors were installed under the grassland. The model can capture the SWC dynamics in response to each rainfall event (Fig. 5a: $R^2 = 0.87$; RMSE = 0.01 $m^3$ $m^{-3}$). In addition, the diurnal patterns were also captured though their amplitudes were not as significant as observations. The simulated Ts at 10 cm depth also displayed an apparent diurnal pattern
(Fig. 5b: $R^2 = 0.86$; RSME = 2.72 °C; Fig. 5c). Overall, the error ranges for model validation were at a reasonable order of magnitude (Table S5) as reported in other studies in Yanchi County (Gong et al., 2016; Du et al., 2021). The simulations of GPP and LE exhibited the most substantial discrepancies among all the fluxes. A thorough analysis of these deviations will be discussed in Section 4.1.



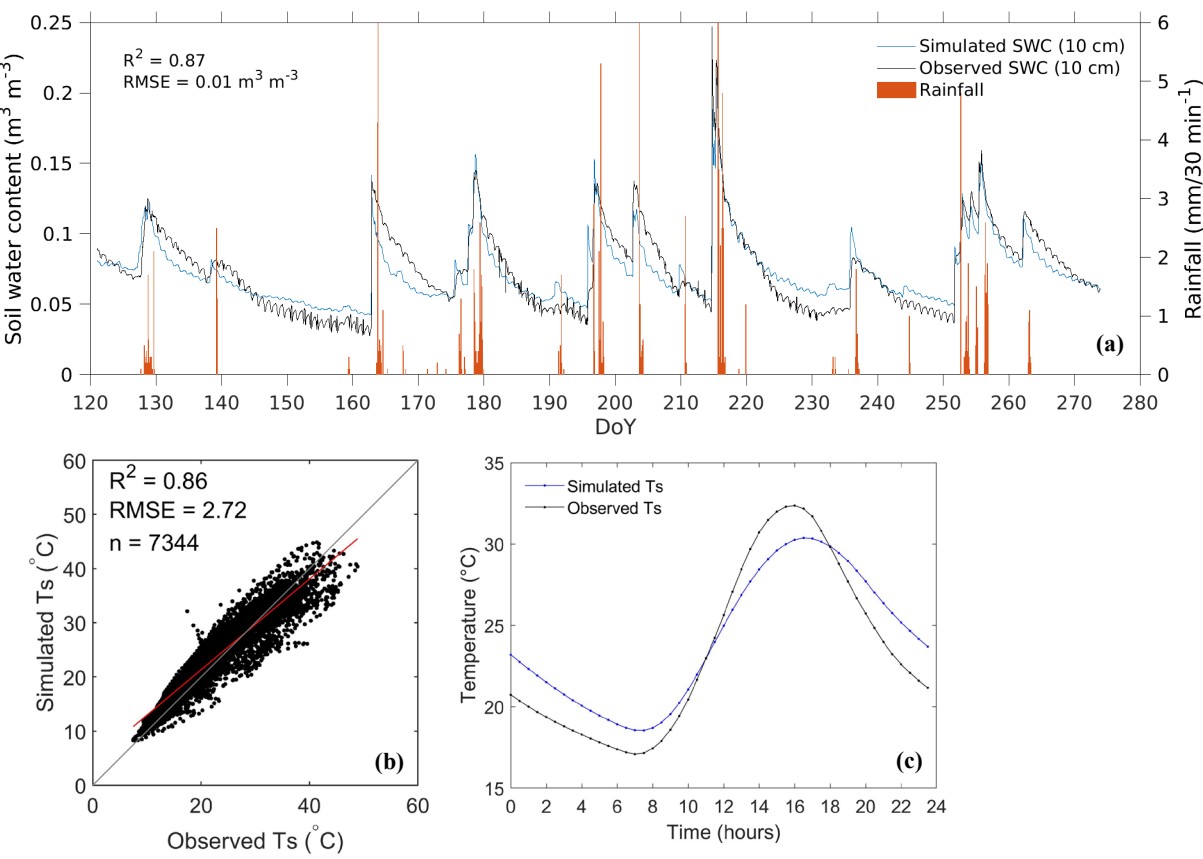

**Figure 5. Model simulated and measured half-hourly soil water content (SWC) and (b) soil temperature (Ts) during May-September in 2019; (a) Temporal dynamics of SWC; (b) Scatter plots between the simulated and the observed Ts; (c) Diurnal cycle of simulated and observed Ts.**

### 3.3 Comparison between two scenarios

The averaged values of simulated fluxes during May-September in 2016 and 2019 were used to understand the diurnal and daily variations. As shown in Table S6-S8, the differences in fluxes between two ecosystems were calculated by subtracting the fluxes of the shrubs-grassland ecosystem from that of the grassland ecosystem. Averaged values and seasonal totals were expressed using mean ($\pm$ standard deviations) in the tables.

### 3.3.1 Diurnal variations of energy fluxes

The envelope of net radiation (Rn) ranged from $-91.54$ W m$^{-2}$ to $506.82$ W m$^{-2}$ (Fig. S9 (a)). The shrubs-grassland ecosystem was likely to receive more radiance because of the denser canopies of shrubs, reflected by the larger LAI in the model.

However, differences in diurnal Rn between the two ecosystems were very small and mainly occurred at midday (i.e., 10:00 - 15:00 h), with an averaged difference of 24.78 $\pm$ 4.49 W m$^{-2}$. At night hours (i.e., 17:00 - 07:00 h), the Rn were almost the

same for two ecosystems, with the negative values ($\sim -40$ W m$^{-2}$) induced by the outgoing longwave radiation from soil and leaves.

The sensible heat flux (H) followed a similar pattern like the Rn but with a larger difference between the two ecosystems. The H reached the peak at ~262 (204) W m$^{-2}$ in May and July of the shrubs-grassland (grassland) ecosystem (Fig. S9 (b)). During

nighttime, H was below zero, indicating a heat transfer from the atmosphere to the ground due to the lower surface temperature of soil and canopy. The H of shrubs-grassland ecosystem appeared to be larger than that of grassland ecosystem whereas the seasonal H/Rn partitioning was similar ($\sim 37$ %) for both ecosystems.

Latent heat flux (LE) had the maximum diurnal peak at 158.72 W m$^{-2}$ (143.38 W m$^{-2}$) in August of shrub-grassland (grassland)

ecosystem (Fig. S9 (c)). The LE in both ecosystems steadily increased from June to August and dropped in September, which was in line with the plants growing stages and rainfall pattern. The seasonal LE/Rn ratio of the shrubs-grassland ecosystem (52 %) was greater than that of the grassland ecosystem (49 %).

During May-September, the ground heat flux (G) peaked at $\sim 11:00$ am, with the averaged values of 129.86 $\pm$ 5.88 W m$^{-2}$

(159.57 $\pm$ 7.74 W m$^{-2}$) for the shrubs-grassland (grassland) ecosystem (Fig. S9 (d)). More heat is transported through the surface and soil in the grassland ecosystem because less vegetation coverage induces more energy exchange in the soil. Throughout the growing seasons, more energy was stored in the soil under the grassland ecosystem (13.79 W m$^{-2}$) than that under the shrubs-grassland ecosystem (9.96 W m$^{-2}$).

**3.3.2 Daily variations of water fluxes**

We compared daily variations of soil water content (SWC), evaporation and transpiration between the grassland ecosystem and shrubs-grassland ecosystem, respectively. To see how the rainfall availability influenced the water fluxes, the SWC, E and T were also compared between year 2016 and 2019. In this study, we defined the year 2016 (2019) with seasonal rainfall of 218.1 mm (292.4 mm) as a relatively *dry* (*normal*) year, referring to the mean annual precipitation (i.e., 296.99 mm) as a

baseline.

*Water fluxes: SWC*

Generally, the SWC decreased in every soil layer after planting shrubs (Fig. 6). Significant decreases in SWC ($\sim 24$ %) occurred between 50 cm depth to 100 cm depth, where most of the fine roots of shrubs concentrate (Jia et al., 2012; Zhu and Wang,



2020). The SWC in the effective root zone (i.e., 0-200 cm) decreased by 19 % (Eq. S15). Out of the root zone (i.e., below the

250 cm depth), the changes in SWC became less variable.

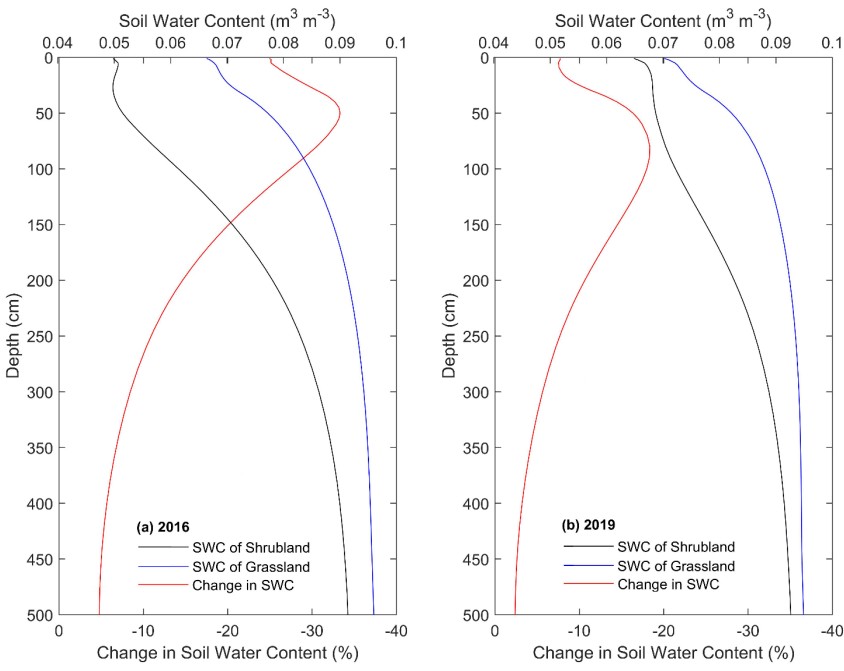

**Figure 6. Profile of the changes in soil water content (SWC) after planting shrubs, which is calculated by using averaged SWC under the shrubland (black line) minus averaged SWC under the grassland (blue line) over the growing season (May to September) in (a)**
**2016, (b) 2019. Noted that the comparison was carried out between grassland ecosystem and shrubland ecosystem, instead of shrubs-grassland ecosystem.**

The SWC at 10 cm soil depth (hereafter denoted as surface SWC) under grassland peaked at ~ 0.15 $m^3$ $m^{-3}$ in July and August,

followed by the most frequent rainfall events during the year (Fig. 7a). We note that the surface SWC under the two land

covers was highly responsive to the rainfall events, represented as abrupt increases even within several hours (Fig. 7a). At 100

cm soil depth (Fig. 7b), the SWC was less influenced by water exchange between the soil and atmosphere (i.e., evaporation

and precipitation) (Zeng et al., 2009a; Yang and Fu, 2017; Zeng et al., 2009b).

The SWC under the shrubland was generally smaller than under the grassland. Moreover, the difference in averaged SWC

between two ecosystems at 100 cm depth with the value of 0.023 (0.016) $m^3$ $m^{-3}$ was much greater than that at surface SWC

with the values of 0.017 (0.006) $m^3$ $m^{-3}$ in 2016 (2019) (Table S6). These contrasts indicate that the revegetation decreased the

SWC and had a more significant effect on the SWC at deep soil layers. Particularly, in the dry year 2016, the differences in

SWC between two land covers were more prominent and the shrubs tended to consume more soil water than in 2019 (Fig. 7:

left against right). These observations reflect that shrubs can utilize the soil water at deeper soil layers with their developed

root systems when experiencing drought (Wang et al., 2018; Zhu and Wang, 2020; Zhang et al., 2020b).






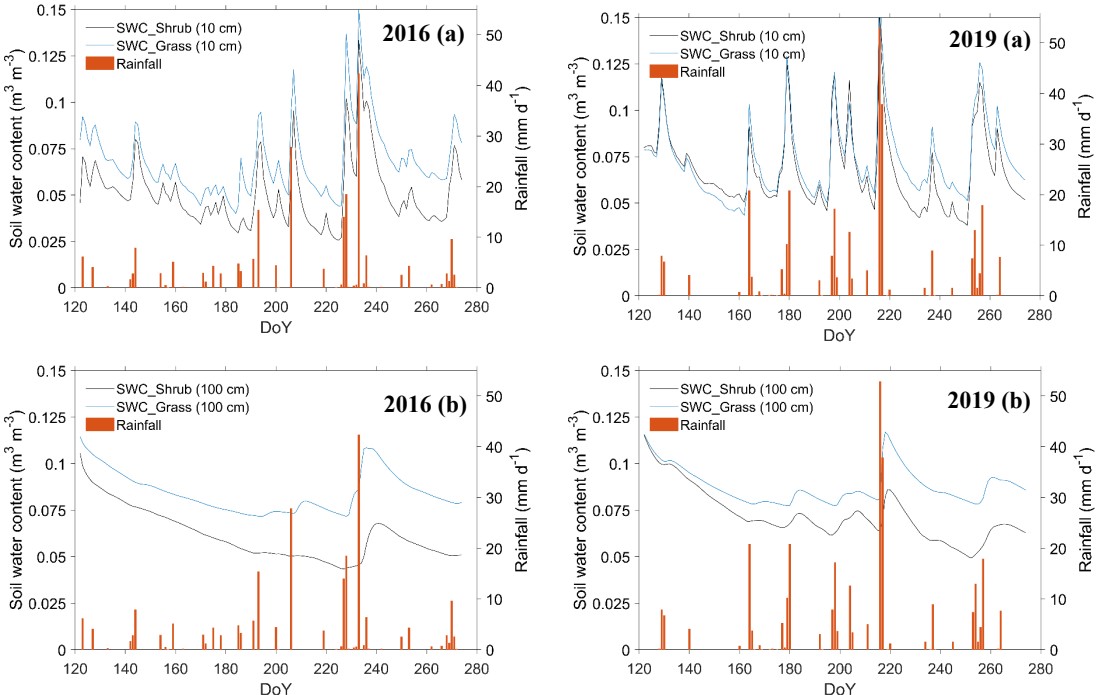

**Figure 7. Daily variation of simulated SWC at (a) 10 cm and (b) 100 cm depth under two land covers during May–September in year 2016 (left) and 2019 (right). In the legend, 'shrub' denotes the shrub grid modelling.**

### *Water fluxes: Evaporation, Transpiration and ET*

As shown in Fig. 8, the evaporation and transpiration boosted along with the rainfall pulses and their large day to day variation was noted. The seasonal evaporation of the shrubs-grassland ecosystem was reduced by 30 % (16 %) compared to that of the grassland ecosystem over the growing season in 2016 (2019) (Table S7). In contrast, the seasonal transpiration of the shrubs-grassland ecosystem was increased by 87 % (111 %) compared to that of the grassland ecosystem in 2016 (2019). Over the

growing season in 2016 (2019), transpiration accounted for 54 % (49 %) of ET from the shrubs-grassland ecosystem while only accounted for 30 % (28 %) of the ET from grassland ecosystem. The seasonal ET of the shrubs-grassland ecosystem increased by 6 % (20 %) compared to that of the grassland ecosystem in 2016 (2019). In conclusion, the revegetated shrub reduced the evaporation but enhanced the transpiration, ultimately increasing the total ET at the ecosystem level.

As rainfall increased by 74.3 mm in 2019, the seasonal ET increased by 59.50 mm (20.88 mm) of the shrubs-grassland (grassland) ecosystem. However, the contributions of evaporation (~ 71 %) and transpiration (~ 29 %) to total ET remained stable for grassland ecosystem in two years. Besides, a slight variation was observed in SWC at 10 cm and 100 cm soil depth under grassland between two years (Table S7). It seems possible that the water consumption of grassland was relatively stable regardless of changes in precipitation amount (see discussion in Section 4.3).





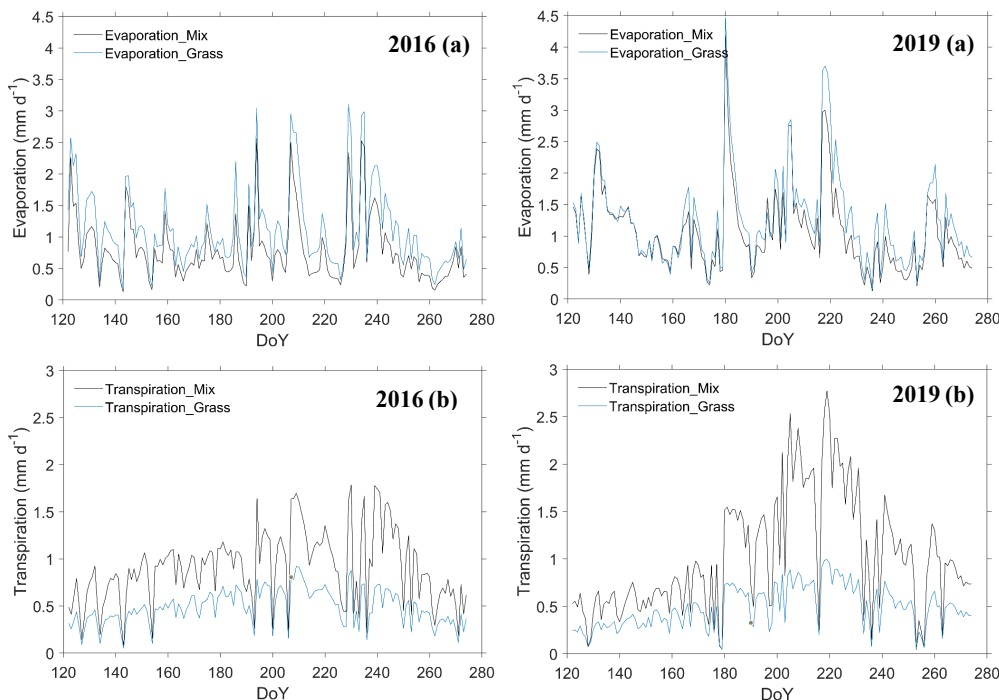

**Figure 8. Daily variation of simulated (a) Evaporation and (b) Transpiration of two ecosystems during May–September in year 2016 (left) and 2019 (right).**

### 3.3.3 Diurnal and daily variations GPP

The Gross Primary Productivity (GPP) of both ecosystems displayed evident diurnal patterns (Fig. S10 (a)). During the daytime, GPP was positive, indicating that the plants were taking up carbon. The plants halted their photosynthesis at nighttime with zero value of GPP. The magnitude of the diurnal variations was the largest in August, during which the GPP peaked at 7.13 $\mu mol\ m^{-2}\ s^{-1}$ and 3.49 $\mu mol\ m^{-2}\ s^{-1}$ during midday (10:00-14:00 h) of shrubs-grassland and grassland ecosystems, respectively. The GPP of the shrubs-grassland ecosystem was significantly larger than grassland and such difference was much more significant in July and August.

Interestingly, the GPP of the grassland ecosystem reached a plateau at ~10:00 am and then dropped at midday, particularly in July (Fig. S10 (a)). The likely cause is the midday depression phenomenon that high radiation induces the saturation of the canopy photosynthesis, which declines the stomatal conductance (Chen et al., 2014; Wang et al., 2019b). The simulations show the midday depression is less evident in the shrubs (Fig. S10 (b)), which is also evidenced by a field study on the transpiration characteristics of *Caragana intermedia* in the Mu Us Sandy Land (Zang et al., 2009).




Daily GPP of the shrubs-grassland ecosystem tended to be more sensitive to the rainfall with more significant fluctuations, represented as the apparent increase in the DOY 207, 229 and 234 in year 2016 and DOY 180, 199, 217 in year 2019 (Fig. 9). Moreover, the shrubs-grassland ecosystem assimilated 69 % (96 %) more carbon than the grassland ecosystem over the

350 growing season in 2016 (2019) (Table S8). In 2019, the seasonal GPP of shrubs-grassland ecosystem (grassland ecosystem) increased by 24 % (6 %) compared to that in 2016. In the presence of more rainfall, the shrubs suffered less water stress, directly promoting the carboxylation rate for greater photosynthesis (Eq. S18).

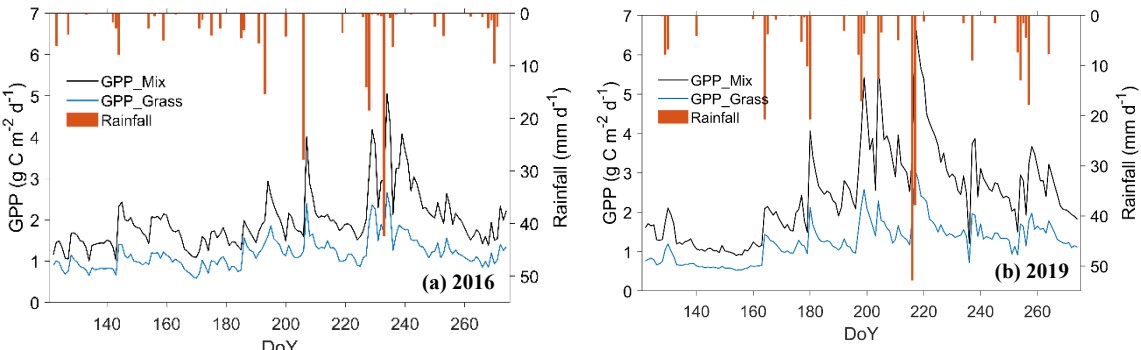

**Figure 9. Daily variation of simulated Gross Primary Productivity (GPP) of two ecosystems during May–September in (a) 2016 and**
355 **(b) 2019.**

### 3.3.4 Changes caused by revegetation

Overall, our results show that revegetation increased Rn, LE and H by 5 %, 13 % and 8 %, respectively, while decreasing G by 28 %. The root zone water at 0-200 cm soil depth was reduced by 19 %; meanwhile, the transpiration increased sharply by
360 99 %. As for carbon flux, the revegetated shrubs increased the GPP by 82 % (Fig. 10).

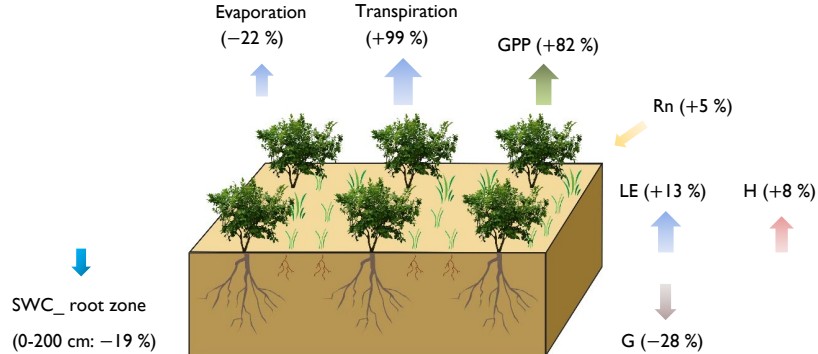

**Figure 10. The changes in the energy, water and carbon fluxes caused by the revegetation, illustrated by the difference in averaged fluxes between shrubs-grassland ecosystem and grassland ecosystem. The arrows indicate the direction of the fluxes and the symbol '−' represents a decrease while the '+' represents an increase based on the grassland scenario.**



## 4 Discussion

### 4.1 Evaluation of model performance and uncertainties

As summarized in Table S5, the model simulations during calibration and validation showed a good match with the observed fluxes, but still, there are some deviations. Firstly, these deviations primarily stem from the model inputs and the scenarios design. For instance, (i) The estimated parameters (e.g., rooting depth and $V_{cmax}$) and reconstructed LAI for shrubland and grassland were critical for LE and GPP modelling; (ii) Concerning the scenarios design, the contributions from shrubland and grassland that estimated from only one UAV photo might not be representative over the investigation period because vegetation coverage is highly dependent on precipitation supply in the study area. The contributions from shrubland and grassland implicitly included the fraction of bare soil. The model then simulates evaporation from the soil beneath sunlit or shaded leaves and upscales it into a canopy/grid by multiplying LAI, rather than directly calculating evaporation from the bare soil. However, this approach may lead to an underestimation of soil evaporation, as a separate simulation of bare soil was not conducted due to the distortion of the simulations with zero value of LAI; (iii) The quality of gap-filled forcing data and in-situ measurements are the basis for a valid comparison between simulations and observations. Although we applied a pre-assessment and filtering strategy in the energy fluxes (Eq. S10), the energy closure issue results in uncertainty in comparing the LE/ET values.

For the model itself, it is a one-dimensional vertical model without considering the lateral flow, which is assumed reasonably to be negligible in this flat study area. Nevertheless, at another site in Yanchi County with similar land covers, Gong et al. (2016) found that the absence of horizontal exchanges of water, vapour and heat advection in the model led to an underestimation of energy fluxes. However, there is always a trade-off between model complexity and effectiveness. And for our focus on understanding the impact of revegetation on energy, water and carbon fluxes, the neglect of lateral flow in the STEMMUS-SCOPE model will not affect our conclusions.

Validation performances of shrubs-grassland ecosystem LE and GPP ($R^2$ = 0.66 and 0.68) were reasonable. Most of the discrepancies are represented as the overestimate of LE during large and continuous rainfall events. For instance, in 2016, the model generated outliers during DOY 228 - 240 along with 69.56 mm rainfall within this period; likewise, the same was also reflected in 2017 and 2019 (Fig. S11). The overestimation of energy-limited evaporation rate, driven by the skin temperature, is the main reason for the overestimation of LE. The causes of the outliers might be attributed to two factors: First, the use of a fixed time step (i.e., 30 minutes) in running STEMMUS-SCOPE was too coarse temporally to achieve a precise numerical solution. Alternatively, the measurements of LE during rainfall events are less reliable. During the dry period without rainfall, the model tended to underestimate LE.

As shown in Fig. S12, GPP simulations were underestimated during the water-stressed period (i.e., water stress factor (WSF) close to zero). We note that GPP simulation is positively correlated to the variations of WSF, which directly regulates the



$V_{cmax}$ thereby influencing GPP (Eq. S18). Several factors are known to be partially responsible for the underestimations in LE and GPP. First, limited information on soil moisture in the deep soil layer could contribute to these underestimations
(Valayamkunnath et al., 2018). In the STEMMUS-SCOPE model, the initial SWC profile not only determines the pattern of soil water storage in the soil column but is also an indication of the pattern of root water uptake. The underestimation of LE and GPP during the water-stressed period suggests that the wetter deep soil layers might exist for sustaining plant growth but were not captured by the simulation. Second, the approximated LAI for grassland (Section 2.3.2) and estimated $V_{cmax}$ might not be representative of the field conditions. Our results suggest that accurate information on LAI, rooting depth and soil
moisture in multiple layers is required to improve the model predictions.

## 4.2 Effects of anthropogenic revegetation on ecosystem processes

### 4.2.1 On energy fluxes

The difference (5.17 ± 12.87 W m$^{-2}$) in net radiation (Rn) is insignificant between the two ecosystems (Fig. S9 (a)) because
the same meteorological forcings (mainly referring to downward shortwave and longwave radiation) were used for modelling. The averaged midday (i.e., 11:00 am) ground heat flux (G) accounted for 30 % (39 %) of Rn in the shrubs-grassland (grassland) ecosystem (Fig. S9). The observed importance of G was also reported in other semiarid and arid ecosystems with a dry soil surface and low vegetation coverage (Jia et al., 2016; Heusinkveld et al., 2004; Purdy et al., 2016; Kurc and Small, 2004).

The ratio of averaged latent heat flux (LE) and Rn over two growing seasons was 52 % (49 %) of shrubs-grassland (grassland) ecosystems whereas, indicating that turbulent energy was dominated by LE in this semiarid region. The transition from LE to sensible heat flux (H) dominance occurred from ~8:00 am to ~16:00 pm, during which the relatively high air temperature was a significant factor to affect H. Compared to the grassland ecosystem, the revegetated shrubs increased the H and LE. Similar findings have also been reported for a semiarid watershed in southeastern Arizona (Flerchinger et al., 1998), three semiarid
ecosystems (cheatgrass, sagebrush and lodgepole pine) in the Snake River basin (Valayamkunnath et al., 2018) and a shrub-steppe ecosystem in Yanchi County (Gong et al., 2016). The consensus was that LE is positively correlated to the LAI, vegetation coverage and water availability. The H is positively correlated to the surface temperature.

### 4.2.2 On gross primary productivity

Despite the underestimation of simulated GPP, the diurnal and monthly variations were in line with observations or simulations in other semiarid shrubs-grassland ecosystems in China (Jia et al., 2018; Du et al., 2021; Ma et al., 2020). It has commonly been concluded that revegetation enhanced carbon assimilation at the ecosystem level. Such an enhancement in this study (i.e., the difference in GPP between the two ecosystems) was more pronounced in 2019, which received more precipitation (Table



S8). Besides, the significant carbon uptake in both ecosystems was noticed after each rainfall event (Fig. 9). This rainfall
dependency of carbon flux is a representative characteristic in the semiarid regions, where the water and carbon cycles are
tightly-coupled (Silva et al., 2017; Brümmer et al., 2008; Hastings et al., 2005; Eamus et al., 2013). Many studies reported that
the shrublands are a stronger net carbon sink than the C3 grasslands (Eamus et al., 2013; Zhang et al., 2020a; Hastings et al.,
2005; Petrie et al., 2015).

### 4.2.3 On water fluxes

The simulated and observed SWC values at 10 cm soil depth agreed well during the calibration ($R^2$=0.89; RMSE=0.01 $m^3$ $m^{-3}$) and validation stage ($R^2$=0.87; RMSE=0.01 $m^3$ $m^{-3}$). Rather than the daily values of SWC, the instantaneous values at half-
hour time steps were captured by the STEMMUS-SCOPE model, illustrating the timely SWC responses to rainfall events.
This can facilitate the future investigation of the responses of plants (e.g., stomatal conductance and leaf water potential) to
the water deficit (Liu and Shao, 2015; Fang et al., 2011; Du et al., 2021). The rapid uptake of surface layer soil water
replenished by rainwater was consistent with that observed in *Caragana intermedia* plantation in the northeast Tibetan Plateau
(Zhu and Wang, 2020).

Furthermore, our simulations indicated that SWC decreased within the 0-500 cm profile after revegetation. The most
significant decrease occurred at 5-200 cm soil depth, which was highly associated with root distribution and rooting depth of
the shrubs. The shrubs extracted more soil water than grasses, especially water from the deep soil. Moreover, such extraction
was more intense in the drought year. Similarly, the simulations from the SHAW model suggested that SWC under
the *Caragana korshinskii* decreased within the 1.0-4.0 m profile and the SWC was depleted from deeper soil with the
development of dry soil layers below 1.0 m (Liu and Shao, 2015). Our findings are also supported by field investigations on
the water uptake patterns of shrubs in the semiarid steppe of northern China, using stable isotopes technique or installing the
sensors in multiple soil layers (Wang et al., 2018; Zhu and Wang, 2020; Zhang et al., 2020b; Jia et al., 2012; Wang et al.,
2019a; Jian et al., 2015). These experiments illustrated that the shrubs with a deep-rooted system, such as *Caragana* species,
can flexibly switch their water source to deeper soil layers when the soil water in the shallow layers is depleted.

The seasonal evapotranspiration (ET) of the shrubs-grassland ecosystem was higher than that of the grassland ecosystem,
although the evaporation of the shrubs-grassland ecosystem was lower (Fig. 8 and Table S7). The revegetated shrubs have
increased the LAI, directly diminishing the energy that reached the soil surface and thus decreasing evaporation. In this sense,
revegetation increased the ecosystem ET mainly with the increasing transpiration (i.e., root water uptake). Our conclusions are
qualitatively consistent with other studies in the same study area (Du et al., 2021; Dan et al., 2020), a shrub-encroached steppe
ecosystem in Inner Mongolia (Wang et al., 2018), and a modeling study on the water-energy balance of shrubland-interspace
in Yanchi County (Gong et al., 2016). However, the ET variations between two ecosystems in our study differed from findings
from Kurc and Small (2004 and 2007), who reported that ET time-series were similar at a grassland and shrubland site. Given



the similar semiarid climate condition, the most likely causes of the different effects of shrubs on ET components are the difference in plant traits (e.g., vegetation coverage and rooting depth) and soil properties (e.g., soil texture, $K_{sat}$, VG parameters).


An interesting finding is that the shrubs-grassland ecosystem presented a higher ratio between transpiration and evapotranspiration (T/ET) in the *dry* year 2016 (54 %) than that in *normal* year 2019 (49 %) (Table S7). This observation implies that the surplus rainwater in the wetter year was not absorbed effectively by roots but was evaporated or recharged deep soils (> 300 cm) (Kurc and Small, 2007, 2004; Gao et al., 2023). Similarly, Chen et al. (2014) found that concentrated

rainfall events did not induce a significant increase in transpiration of two revegetation species unless the rainwater could infiltrate the deep soils. At a semiarid loess site vegetated by apple trees in China's Loess Plateau, Gao et al. (2023) also observed a higher T/ET ratio during the extremely dry years as compared to extremely wet years; they also noted that the relationship between T/ET and annual precipitation could be complex (either independent or negatively correlated) and may be influenced by local soil condition (e.g., initial soil wetness, soil hydraulic properties). The intricate interplay between T/ET

and annual precipitation of thick and dry loess soil profiles in the context of climate change presents a challenging topic.

### 4.3 Will the revegetated shrubs lead to soil water depletion?

For the grassland ecosystem, the ET partitioning, SWC at 10 cm and 100 cm depth and GPP remained relatively stable in two years. Additionally, planting shrubs decreased the SWC and enhanced the water losses. Consequently, we hypothesized that

the water consumption of shrubs-grassland ecosystem might reach its limit. To examine this assumption, we further compared the cumulative rainfall and ET in Fig. 11.

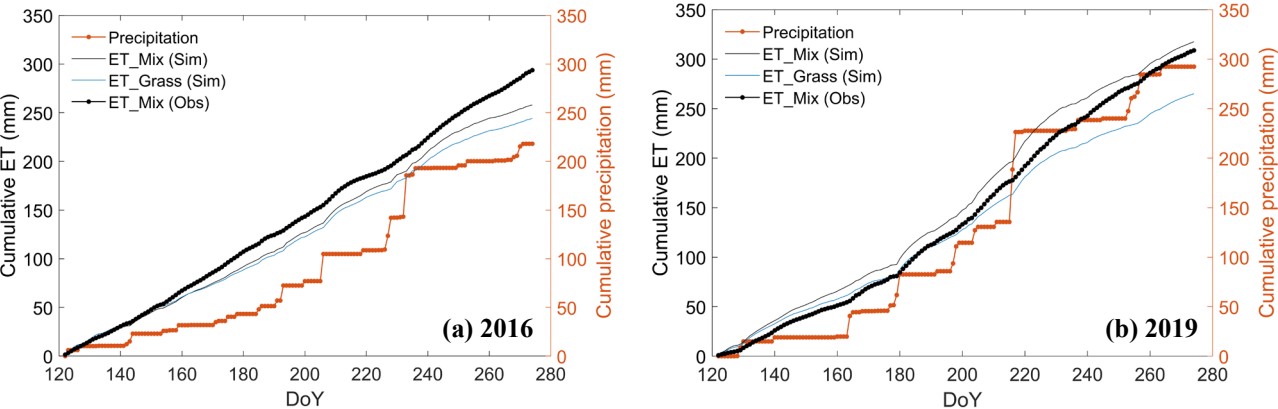

**Figure 11. Comparison of cumulative precipitation and cumulative ET of two ecosystems over the growing season in year (a) 2016 (*dry*) and (b) 2019 (*normal*), where the 'ET_Mix (Sim)' and 'ET_Grass (Sim)' are the simulated ET for shrubs-grassland and**
**grassland ecosystem, respectively; and the 'ET_Mix (Obs)' is the observed ET (i.e., LE) from EC tower.**



The year 2016 witnessed an excess of 40 mm (76 mm) in cumulative simulated (observed) ET over the received precipitation (Fig. 11a). This observation points to an additional water source needed for ET, which can only be attributed to water transpired from deeper soil layers. In the wetter year 2019 (Fig. 11b), the rainwater replenishment seems enough to sustain the growth of grasses but still not enough for the shrubs-grassland ecosystem. Cumulative simulated (observed) ET exceeded the

precipitation by 25.19 mm (16.43 mm) in 2019. In comparison, the higher excess of ET over precipitation in 2016 provides evidence that drought conditions could result in a more significant depletion of water from deeper soil layers. Such excessive water consumption could lead to soil desiccation as reported in other semiarid regions with revegetation practices in China Loess Plateau (see supporting information Table S1 of Zhang et al., 2018). However, here it is only an implication based on simulations over two growing seasons. Scenario simulations for longer years are required to understand the environmental

limit of explorable water resources in the study area.

**5 Conclusion**

To understand the effects of revegetation on eco-hydrological processes of a desert steppe in northwestern China, we simulated the energy, water and carbon fluxes during May-September in 2016 (*dry*) and 2019 (*normal*), for a shrubs-grassland scenario and a grassland scenario, respectively. Simulations for two land covers were driven by corresponding LAI time-series and

plant traits parameters in the STEMMUS-SCOPE model. Simulated fluxes based on half-hourly time steps agreed well with the measured trends. In particular, the model can simulate the soil water content accurately and capture its diurnal and daily dynamics. According to the results of comparison between the two scenarios, the revegetation practices in the study area (i) increased the latent heat flux and sensible heat flux and decreased the ground heat flux, in which the latent heat flux dominated the energy partitioning; (ii) promoted gross primary productivity, which was highly responsive to rainfall availability; (iii)

decreased soil water content at 0-500 cm soil depth (especially 50-200 cm) via root water uptake, which was more pronounced during the drier year; (iv) aggravated the water consumption of ecosystem with the decrease in root zone water and remarkable increase in transpiration. Moreover, revegetated shrubs have disrupted the water balance, manifested by greater evapotranspiration than received precipitation in two growing seasons. Future revegetation practices should take into account the sustainable limits of ecosystems to avoid soil water depletion, which risks triggering the imbalance of the tightly-coupled

energy, water and carbon cycles in the arid and semiarid regions.



*Code and data availability.* The input data, source code and output data of STEMMUS-SCOPE are available on Zenodo at
https://doi.org/10.5281/zenodo.7986566.

*Author contributions.* ET, Z.Su and YZ designed and performed the study. ET ran the simulations, analysed the data and prepared the original draft. LD, HW and CQ offered the field data. YW, Z.Song, DY and CvdT provided technical help in simulations and data analysis. All authors reviewed and edited the manuscript.

*Competing interests.* The authors declare that they have no conflict of interest.

*Acknowledgements.* This research has been funded by The Netherlands Organisation for Scientific Research (NWO) KIC, WUNDER project (grant no. KICH1. LWV02.20.004), Netherlands eScience Center, EcoExtreML project (grant ID. 27020G07) and Water JPI project "iAqueduct" (Project number: ENWWW.2018.5). In addition, this study was supported in part by the ESA ELBARA-II/III Loan Agreement EOP-SM/2895/TC-tc and the ESA MOST Dragon IV Program. We thank the National Natural Science Foundation of China (grant no. 41971033 and 41967027), Fundamental Research Funds for the Central Universities, CHD (grant no. 300102298307) and Ningxia Province's Natural Science Foundation (grant no. 2022AAC02011) for data support.

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
