# Peer review of "Understanding the Effects of Revegetated Shrubs on Fluxes of Energy, Water and Gross Primary Productivity in a Desert Steppe Ecosystem Using STEMMUS–SCOPE Model"

_Biogeosciences, 2023_

## Author Comment (AC1)

**Response to reviewer #2**

**General comments**

*The authors used the STEMMUS-SCOPE Model to analyze how the revegetation of shrubs in a desert steppe modifies the water, energy and GPP dynamics during wet and dry years. Since the paper is not a model development study, I will focus on its presentation aspects. Nonetheless, I think the paper is quite straightforward, but can be improved by taking care of my suggestions.*

Thanks for the kind words! Your comments are immensely valuable in enhancing the manuscript's quality.

*The title: Since the only carbon flux analyzed is GPP, I recommend the title be changed to "Understanding the Effect of Revegetated Shrubs on fluxes of Energy, Water and Gross Primary productivity in a Desert Steppe Ecosystem Using STEMMUS-SCOPE Model". Additionally, if the EC tower measured $CO_2$, how about also showing the net ecosystem exchange as one indicating effect of different vegetation? Or it is not possible to differentiate the vegetation effect from NEE?*

We agree with the title modification because we only presented the simulation of Gross Primary Productivity (GPP). It's true that the EC tower measured Net Ecosystem Exchange (NEE) and the GPP was partitioned from the measured NEE. In contrast, the model usually calculates photosynthesis (i.e., GPP) first and then determines the Net Ecosystem Exchange (NEE) by subtracting ecosystem respiration (Reco) from GPP. In STEMMUS-SCOPE, the Reco is calculated using a simple linear equation, Reco = 2.5 + 0.054375*$T_{surface}$, where $T_{surface}$ represents the surface temperature of leaves or soil. To avoid error propagation from the calculation of Reco in the model, we decided to directly compare GPP, making the comparison more direct and accurate.

**Specific comments**

*Comment 1: Equation (1), what does C_shrub mean?*

**Response 1:** Thank you for your attentive review. We have added an explanation of this variable in the revised manuscript.

$$GPP = C_{shrub}\, GPP_{shrub} + (1 - C_{shrub})\, GPP_{grass} \tag{1}$$

$C_{shrub}$ is the contribution of shrubland to the overall flux, and its value is 58.33 % while the contribution of grassland is $(1 - C_{shrub}) = 41.67$ % (Table 1, Line 125).

*Comment 2: From the method description, it appears LAI is used as model input. Does this mean the model actually does not simulate the carbon cycle, aka it is a prescribed phenology simulation?*

**Response 2:** The STEMMUS-SCOPE model uses leaf area index (LAI) as an input and focuses on carbon assimilation, excluding carbon allocation. We are aware of this issue and will address it in our upcoming study by coupling the STEMMUS-SCOPE model with the WOFOST crop growth model. This coupling will allow us to simulate LAI and plant growth processes effectively. The evaluation of the coupled STEMMUS-SCOPE-WOFOST model, performed with various plant functional types, showed successful simulation of vegetative dynamics, including LAI and plant height. We intend to submit the study of the STEMMUS-SCOPE-WOFOST model to a journal soon.

However, in this manuscript, our primary focus lies on simulating ecosystem flux, and we use MODIS LAI as a reference due to its availability and relatively high temporal resolution. The MODIS LAI proved more sensitive to dense vegetation and might not be representative in the arid region with sparse vegetation (Fensholt et al., 2004; Fensholt and Sandholt, 2005). Therefore, we have corrected the MODIS LAI using field observations and relevant literature values, in order to ensure this critical input as representative as possible.

*Comment 3: Figure 2, what is the uncertainty of the observed LAI? Can you also show it in the Figure?*

[Figure]

**Figure 1. Reconstructed LAI of shrubland and grassland from 2016 to 2019.**

**Response 3:** Apologies for any confusion in the manuscript. The 'observed' LAI values are not subject to uncertainty because they are 'dummy' LAI. The red dots (Obs_LAI_Shrub) are 'observed' LAI of shrubs on DOY 160.5 and DOY 168.5, where we did not have real measurement, but we calculated them using a linear relationship derived from the measurements and MODIS LAI in 2022 (Table S4 in the Supplement). We will modify the caption of Figure 2 to make this clarification.

*Comment 4: 2.4.1 Sensitivity analysis, it was mentioned that SA was only done for parameters of the shrubland simulation, why is it not done for grassland as well?*

**Response 4:** In *Section 2.4.1*, we conducted a sensitivity analysis by running 160 sets of parameters for the shrubland simulation, while a fixed parametrization was used for the grassland simulation. We chose to perform the sensitivity analysis only for shrubland due to the following reasons: First, the primary objective of the sensitivity analysis was to evaluate the impact of parameters on the dominant land cover, which in our case is shrubs. Second, running the model for one set of data would take approximately 40 minutes, making it computationally intensive for 160 sets. Therefore, to optimize computational resources and focus on the primary objective, the sensitivity analysis was limited to the shrubland simulation.

*Comment 5: 3.2 Model performance, in no place I found model spin up was mentioned. Do the model simulations include model spinup? How is it done?*

**Response 5:** No spin-up was performed in the STEMMUS-SCOPE model simulation. Spin-up is typically used to initialize the model with stable initial conditions. Generally, STEMMUS-SCOPE utilized observed soil moisture and soil temperature as initial conditions, which proved to be effective and stable for initializing the simulation (Wang et al., 2021). Additionally, considering computational efficiency and time constraints, we opted not to conduct a model spin-up for this investigation.

*Comment 6: Figure 4c, there are a few points aligning as a vertical line. Does it indicate there are some problems with the data or the model?*

**Response 6:** Figure 4 (c) on the right-hand side compares the observed sensible heat flux (H_obs) and simulated sensible heat flux (H_sim). The points aligning as a vertical line indicate that the observed H were zero but the simulated H were not. A sensible heat flux of zero means no **net** heat transfer through conduction and convection between the Earth's surface and the atmosphere. Moreover, this only happened in the data of 2017 among year 2016-2019.

[Figure]

**Figure 4 (c)**

We acknowledge that there might be issues with the observed data, particularly in 2017, as we did not filter the data in terms of energy balance closure (Line 65 in the Supplement). The lack of surface ground heat flux data in 2017 made it impossible to conduct an energy balance closure assessment and apply the filtering process for that specific year.

Accordingly, we will clarify and modify *4.1 Discussion Line 367* in the revision. **Improved version of Line 367:** (iii) The quality of gap-filled forcing data and in-situ measurements are the

basis for a valid comparison between simulations and observations. For example, some observed H were zero while the simulated values were not in 2017 (Figure 4c). Although we diligently conducted quality control for EC data and filtered the fluxes based on energy balance closure (Eq. S10 in Supplement), we were unable to filter the data for the years 2016 and 2017 due to the unavailability of surface ground heat flux for energy balance closure assessment.

*Comment 7: Figure 5c, I don't quite understand the diurnal pattern. If you are average over a certain time period, please show the mean and as well as the variability (i.e., standard deviation).*

**Response 7:** The figure on the right-hand side now includes the variability. The data points represent the average values of every half-hour in a day over a certain time period. For instance, the mean (± standard deviation) of simulated soil temperature under grassland (blue line) was 23.19 °C (± 4.43 °C) at 0 o'clock, spanning from DOY 121 to DOY 273 in the year 2019.

[Figure]

*Comment 8: Figure 6, could you also show the variability?*

**Response 8:** The variability is added in the figure below. The soil water content (SWC) is the average value of each soil layer over a certain time period. For example, as shown in Figure (b) below, the mean (± standard deviation) of SWC of grassland was 0.07 m³ m⁻³ (±0.03 m³ m⁻³) at 1 cm depth, over DOY 121 to DOY 273 in the year 2019.

[Figure]

*Comment 9: Section 3.3.3 on GPP, I don't think the interpretation of midday depression is accurate. It may involve factors more than radiation. Perhaps you can show the diurnal pattern of leaf temperature, and temperature dependence of carboxylation as well.*

**Response 9:** Thanks for your hints and please see the diurnal pattern of leaf temperature in the next paragraph. Indeed, the high radiation is more like one of the inducements to the close of stomata, represented as middy depression. Figure S10 in Line 260 of *Supplement* shows midday depression of GPP and stomatal conductance.

[Figure]

**Figure S10. Diurnal courses of simulated (a) Gross Primary Productivity (GPP) and (b) stomatal conductance (gs) of two ecosystems during May–September in 2016 and 2019.**

As shown in the figure below, we observed higher midday leaf temperatures in grasses (Tleaf_grass) compared to shrubs (Tleaf_shrub). The increase in leaf temperature could potentially lead to stomatal closure in the plants to regulate water loss, as supported by previous studies (McDowell et al., 2019; Deans et al., 2020; Chen et al., 2014). To enhance the discussion, we will include this figure in Figure S10 and make corresponding clarifications in *Section 3.3.3 On GPP* regarding the impact of leaf temperature in the revision.

[Figure]

We agree that the temperature dependence of carboxylation plays a role because the leaf temperature at the shaded side is used in calculating Vcmax in STEMMUS-SCOPE (Eq. (S23) in the Supplement).

*Comment 10: Discussion 4.2.1, paragraph 2, why don't you use the Bowen ratio as an indicator, which is likely more informative than the ratio of LE/Rn here.*

**Response 10:** We think this is a great idea, and will implement it in the revision.

*Comment 11: Section 4.2.3, on water fluxes. I am wondering if the model can do a good job in overall water mass balance. I am thinking changing from grass to shrub will lead to difference in column integrated water mass, could you show time series of total water storage in the model? Also, the drainage flux? How far the rainfall infiltration could go? I am then also wondering how sensitive is the model simulation to the represented soil depth, given the gravitational drainage condition is used at the lower boundary. Could you elaborate?*

**Response 11:** This is a very nice conclusive question, and we'll break it down into the following 5 sub-points to respond, in terms of the integrated water storage, infiltration flux, drainage flux, overall water balance closure and lower boundary condition.

1. **Integrated Soil Water Storage**

   We did calculate the integrated soil water storage in the root zone (i.e., 0-200 cm depth) (Eq. S15 in the Supplement). Indeed, the replacement of shrubs decreased root zone water storage in 2016 (-27 %) and 2019 (-11 %), respectively. As you suggested, we made a time-series of total soil water storage in the whole soil column (i.e., 0-5 m depth) (see figure below). Similarly, in both years, the total water storage under shrubland was less than that of grassland.

[Figure]

**2. Infiltration**

In the study area, the precipitation-induced surface runoff is rare after the canopy retention or direct infiltration, and the latter is the major process. In the model, the infiltration flux is calculated as the surface moisture flux remaining after surface runoff has been removed from precipitation (refer to CLM model, Niu et al., 2005; Oleson et al., 2004):

$$Infiltration = Precipitation \times (1 - fmax \times e^{-0.5 \times fover \times \frac{Tot_{Depth}}{100}})$$

where $Precipitation$ is the model input [mm]. $fmax$ is the maximum fractional saturated area, which is the percent of area whose topographic index is larger than or equal to the mean topographic index in a grid cell (Oleson et al., 2004). We extracted $fmax$ (= 0.3694) from a global dataset (Niu et al., 2005; Oleson et al., 2004). $fover$ is a decay factor (= 0.5 m$^{-1}$) (Oleson et al., 2004). $Tot_{Depth}$ is the water table depth (= 5 m). It appears that the infiltration amount is closely related to precipitation, and the infiltration rate is related to the saturated water conductivity (Ks).

**3. Drainage**

Given the gravity drainage as lower boundary condition, the temporal changes of liquid flux at the deepest layer (i.e., layer 54[th] at 5 m depth) indicates the water exchange at the boundary layer. As shown in the below figure, (1) the negative drainage flux indicates the water flow downwards to the boundary layer, with a very small value ranges from 0 to -0.005 [mm 30 min$^{-1}$]; (2) The drainage from the grasses was greater than that from shrubs. These observations are in line with the field situation according to the local expert. In this semi-arid area, characterized by low precipitation and deep groundwater level, there is minimal interaction between precipitation and groundwater.

[Figure]

**4. Water Balance Closure**

The water balance closure was evaluated by comparing soil water storage and the difference between water input (i.e., precipitation) and outflow (i.e., simulated evapotranspiration and drainage):

*Change of Integrated soil water storage = +Precipitation - Evapotranspiration - Drainage*

On the right-hand side, we present an evaluation of water balance closure in the grassland simulation of the days without rain in 2019 (i.e., 118 of 153 days). Good agreement was found with values for the RMSE and the index of agreement (*d*-index) equaling 0.42 mm day$^{-1}$ and 0.88, respectively.

The closer to 1 of the *d*-index, the better the performance of water balance closure. The simulated temporal change in SWC was underestimated. The uncertainties might come from (1) the method to calculate the integrated soil water storage (Summation of SWC over all the layers v.s. Inversion of the water balance equation) (Yu et al., 2016); (2) the bias in the simulated LE (i.e., ET) (See the paragraph at Line 390 in manuscript).

[Figure]

**5. Lower boundary condition**

We used gravity drainage as the lower bottom condition in a 0-5 m soil column because the groundwater level in the study area is quite deep (> 6 m). Besides, STEMMMUS-SCOPE model mainly focuses on simulating the soil-vegetation interactions and hasn't included any groundwater modules. Based on the available information on groundwater level and rooting depth of shrub, we think the settings of 0-5 m soil column and gravity drainage are reasonable. Sorry that by now we don't have the answers to *"how sensitive is the model simulation to soil depth and the bottom conditions"*. We can explore adjusting the soil column depth and modifying boundary conditions to address this question. However, we believe these factors may not be our main focus in comparing simulations for two scenarios. Nevertheless, it remains an interesting question to explore in future applications of the model in diverse study areas with varying groundwater conditions.

**References**

Chen, L., Zhang, Z., Zeppel, M., Liu, C., Guo, J., Zhu, J., Zhang, X., Zhang, J., and Zha, T.: Response of transpiration to rain pulses for two tree species in a semiarid plantation, Int. J. Biometeorol., 58, 1569–1581, https://doi.org/10.1007/S00484-013-0761-9, 2014.

Deans, R. M., Brodribb, T. J., Busch, F. A., and Farquhar, G. D.: Optimization can provide the fundamental link between leaf photosynthesis, gas exchange and water relations, Nat. plants, 6, 1116–1125, https://doi.org/10.1038/S41477-020-00760-6, 2020.

Fensholt, R. and Sandholt, I.: Evaluation of MODIS and NOAA AVHRR vegetation indices with in situ measurements in a semi-arid environment, Int. J. Remote Sens., 26, 2561–2594, https://doi.org/10.1080/01431160500033724, 2005.

Fensholt, R., Sandholt, I., and Rasmussen, M. S.: Evaluation of MODIS LAI, fAPAR and the relation between fAPAR and NDVI in a semi-arid environment using in situ measurements, Remote Sens. Environ., 91, 490–507, https://doi.org/10.1016/j.rse.2004.04.009, 2004.

McDowell, N. G., Brodribb, T. J., and Nardini, A.: Hydraulics in the 21st century, New Phytol., 224, 537–542, https://doi.org/10.1111/nph.16151, 2019.

Niu, G. Y., Yang, Z. L., Dickinson, R. E., and Gulden, L. E.: A simple TOPMODEL-based runoff parameterization (SIMTOP) for use in global climate models, J. Geophys. Res. Atmos., 110, 1–15, https://doi.org/10.1029/2005JD006111, 2005.

Oleson, K., Dai, Y., Bonan, B., Bosilovichm, M., Dickinson, R., Dirmeyer, P., Levis, S., Hoffman, F., Houser, P., Niu, G., Thornton, P., Vertenstein, M., Yang, Z., and Zeng, X.: Technical Description of the Community Land Model (CLM), https://doi.org/10.5065/D6N877R0, 2004.

Wang, Y., Zeng, Y., Yu, L., Yang, P., Van der Tol, C., Yu, Q., Lü, X., Cai, H., and Su, Z.: Integrated modeling of canopy photosynthesis, fluorescence, and the transfer of energy, mass, and momentum in the soil-plant-Atmosphere continuum (STEMMUS-SCOPE v1.0.0), Geosci. Model Dev., 14, 1379–1407, https://doi.org/10.5194/gmd-14-1379-2021, 2021.

Yu, L., Zeng, Y., Su, Z., Cai, H., and Zheng, Z.: The effect of different evapotranspiration methods on portraying soil water dynamics and et partitioning in a semi-arid environment in Northwest China, Hydrol. Earth Syst. Sci., 20, 975–990, https://doi.org/10.5194/HESS-20-975-2016, 2016.

---

## Author Comment (AC2)

**Response to reviewer #1**

**General comments**

*This study applied the STEMMUS-SCOPE model to a typical revegetation plot which consists of shrubs and grass, to simulate the impact of revegetated shrubs on surface fluxes (latent heat flux, sensible heat flux, and GPP) and soil moisture. While the manuscript describes a lot about the comparison between the two scenarios, I more focus on the model and the model configuration, and how this study can contribute to model development or deepen our understanding the effect of revegetated shrubs. In general, I think this part of work is weak.*

Thanks for your thorough review and detailed comments on our article. Indeed, the model configuration is the key to represent the fluxes in mixed vegetated areas. In this study, we employed two sets of parametrizations and ran the model separately for two land covers. The accuracy of the composited fluxes demonstrated the feasibility of distinguishing the LAI using HANTS and fractional vegetation coverage to separate the two land covers *(see Section 4. Performance of model calibration in Supplement)*. With this understanding, our goal is to achieve parallel computation of the two land covers in the next model version, with more consideration on interaction of root growth and root water uptake. Below, we address the specific comments related to the model configuration and interpretation of results.

**Specific comments**

1. *Line 29, in the introduction section, the scientific question is not clear. Generally, the authors thought root water uptake is a critical process in the modeling, and the dynamic root length density for estimating root water uptake is necessary. However, no contents about the root water uptake were presented in this study. What is the impact of dynamic root length density? What is the performance of root water uptake simulation? This is the major limitation of this study.*

**Response 1:** Thank you for the insightful suggestions, which will be addressed in the revised manuscript. We did calculate the root zone water storage in 0-200 cm depth (Eq. S15 in the Supplement). And we found that the replacement of shrubs decreased root zone water storage in 2016 (-27%) and 2019 (-11%), respectively. It is a great idea to show the simulation of root water uptake (see figure on the right side), where we compared the root water uptake (RWU) of shrubs and grasses in 2016 and 2019, respectively.

[Figure]

In general, the RWU of grasses (shrubs) increased from the surface layer and then decreased to zero at 30 cm (200 cm) depth. This pattern is highly related to their maximum rooting depths, which were predefined parameters in our model. Moreover, our model successfully captured hydraulic redistribution, as indicated by negative RWU values in the relatively shallow root zone (Kennedy et al., 2019; Wang et al., 2021). The negative RWU values resulted from the higher root water potential (in absolute value) compared to the soil water potential when the surface was too dry in the study area.

To quantify the change in RWU, we used the RWU of grasses as the control reference, then the Changes in RWU (red line) was calculated by $(RWU\ of\ shrub - RWU\ of\ grass)\ /\ RWU\ of\ grass$. When comparing the Changes in RWU, we noticed that the replacement by shrubs reduced RWU at the 0-30 cm depth but increased RWU at the 30-200 cm depth. This observation aligns with the pre-defined root distribution of shrubs and grasses. However, it is important to note that we currently lack observed data to validate the performance of any root simulation in our study.

Refer to the Comments 7, 8 and 12, we will include the root parameters in the sensitivity analysis and optimize them accordingly. Besides, we will analyze the effects of revegetation on root water uptake in a more detailed manner in the revised manuscript. Besides, we will include the equations of RWU and root growth simulation in the revised supplement.

2. *Line 92. The quality control of flux data was missing. Moreover, how did you calculate GPP?*

**Response 2:** Thank you for your attentive review. If necessary, we will include this section in the supplement. The steps to calculate GPP from the raw EC flux are as follows:

  (1) Pre-processing: The raw flux data was first proceeded with EddyPro software.
  (2) Processing: Quality control was conducted. If the value fall in the range as following, it's the invalid value that was set as NA, otherwise the original value was kept.
     - qc = 2
     - $NEE \le -15$ and $NEE \ge 15$;
     - $LE \le -20$ and $LE \ge 550$;
     - $H \le -60$ and $H \ge 400$;

Take the $CO_2$ flux as an example,

| Timestep (0.5 hour) | Raw data | qc = 0 or 1 | qc = 0 or 1 & $-15 \le NEE \le 15$ |
|---|---|---|---|
| 2016 | 7344 | 6602 | 6496 |
| 2017 | 3314 | 2342 | 2322 |
| 2018 | 4368 | 3986 | 3906 |
| 2019 | 7344 | 5297 | 5159 |

(3) Post-processing: Following quality control, NEE is partitioned into GPP using the REddyProc package in R, which involves u* filtering, flux partitioning, and gap-filling steps. Solar radiation (Rg), air temperature (Tair), relative humidity (Rh), and vapor pressure deficit (VPD) were used in this process. The partitioning principle is based on two relationships: (1) at night, Ecosystem Respiration = Net Ecosystem Productivity = - Net Ecosystem Exchange because Gross Primary Productivity is zero at night; (2) Gross Primary Productivity = Net Ecosystem Productivity - Ecosystem Respiration.

3. *Line 125. How did you determine the contributions for shrubland and grassland?*

**Response 3:** The contributions for shrubland and grassland are determined by their fractional vegetation cover. Based on a high-resolution image taken by unmanned aerial vehicle (Fig. S2), the Supervised Classification Method in ERDAS 2020 was employed to determine the fractional cover of shrubs (35 %), grasses (25 %) and bare soil (40 %). Since STEMMUS-SCOPE considers the soil-root-canopy continuum, we implicitly included the bare soil in either shrub grids (58.33 %) or grass grid (41.67 %). The uncertainties of estimating the fractional cover based on limited image were indicated in Line 370.

4. *Line 136, can 500 m-MODIS LAI represent the 30 m fenced area?*

**Response 4:** Indeed, it might be not representative especially for the sparse and mixed vegetated area (Fensholt et al., 2004; Fensholt and Sandholt, 2005). That's why we tried to reconstruct the MODIS LAI using field observations and relevant literature values, in order to ensure this critical input as representative as possible. We corrected the MODIS LAI by the correlation ratio that was estimated based on relationship between MODIS LAI and field measurement in 2022, and lastly compared the corrected values in 2016-2019 with literature values (Line 150, Figure 2). However, the MODIS LAI was used as reference because of its availability and relatively high temporal resolution.

5. *Line 139-140, how did you determine the values of 2.33 and 1/4?*

**Response 5:** Sorry if this was unclear in the manuscript and this will be clarified in the revised version. First, The MODIS 4-day LAI data during 2016-2019 was smoothed by the Harmonic Analysis of Time Series (HANTS) algorithm (i.e., $LAI_{HANTS}$). Second, the linear relationships were determined between $LAI_{HANTS\_shrub}$ and two observed LAI for shrub ($LAI_{actual\_shrub}$) in 2022. The correlation ratio ($ratio = \frac{LAI_{actual\_shrub}}{LAI_{HANT\_shrub}}$) were determined as 1.92 and 2.73 for DOY 160.5 and DOY 168.5, respectively. The final correction ratio applied in $LAI_{HANTS\_shrub}$ ends with value of 2.33, which is the average of 1.92 and 2.73. For detailed calculations of $LAI_{actual\_shrub}$, please refer to Table S4 in the Supplement.

For the LAI of grasses ($LAI_{grass}$), it was estimated as 1/4 of that of the shrubs ($LAI_{shrub}$) based on the following constraints (Line 140 – Line 145):

    i.     $LAI_{shrub}(i) \approx 4\ LAI_{grass}(i)$ (Dan et al., 2020)

ii.   $LAI_{grass}(i)$ should follow the temporal pattern of $LAI_{MODIS}(i)$ and it was ~0.5 m² m⁻² (Yang et al., 2019; Dan, 2020)

iii.  $f_{shrub} * LAI_{shrub}(i) + f_{grass} * LAI_{grass}(i) + f_{baresoil} * LAI_{baresoil} = LAI_{MODIS}(i)$

iv.   $f_{shrub} + f_{grass} + f_{baresoil} = 1$

v.    $LAI_{baresoil} = 0$

where $f_{shrub}$, $f_{grass}$ and $f_{baresoil}$ are the fractional cover of shrubs (35%), grasses (25%) and bare soil (40%), respectively.

6.  *Line 151, what is the difference between red and yellow dots?*

**Response 6:**

[Figure]

**Line 150   Figure 1. Reconstructed LAI of shrubland and grassland from 2016 to 2019.**

The yellow dots and purple dots are reference values from literature while the red dots are the actual observed value of shrub. In Figure 2 (Line 150), the yellow dots and dotted lines (Ref_LAI_Shrub) represent the ranges of measured LAI of the nearby shrublands from the reference of Dan, 2020. The red dots (Obs_LAI_Shrub) are the actual LAI of shrubs on DOY 160.5 and DOY 168.5, where we did not have real measurement, but we calculated them based on the correlation ratio derived in 2022 (Table S4 in the Supplement).

7.  *Line 176, why were root-related parameters not identified as influential parameters? This is the main focus of your study.*

**Response 7:** Thank you for bringing up this important point! In STEMMUS-SCOPE, the maximum rooting depth, fitted extinction coefficient, and root length density are the primary rooting parameters in determining root distribution, root growth, and root water uptake (Jackson et al., 1997; Wang et al., 2021). In the revised version, we will incorporate these parameters into the sensitivity analysis.

8. *Line 216, how did the authors optimize the parameters (best-fit trail in Line 196)? How to avoid the equifinality for the parameters of shrub and grass?*

**Response 8:** During the sensitivity analysis, 160 sets of parameters for shrubland were generated and while a fixed parametrization was used in grassland simulation. As a result, the shrubland simulation generated 160 sets of fluxes, which were aggregated with the fluxes from the grassland simulation. By comparing the 160 sets of aggregated fluxes with the observed fluxes, we calculated the $R^2$ and RMSE for each flux in each trail. At last, an objective function, the normalized root mean square errors $RMSEn = \frac{RMSE_{SWC}}{\overline{Obs}_{SWC}} + \frac{RMSE_{LE}}{\overline{Obs}_{LE}} + \frac{RMSE_{GPP}}{\overline{Obs}_{GPP}}$ was calculated for each trail, where $\overline{Obs}_{SWC/LE/GPP}$ is the average values of observed SWC, LE and GPP throughout the investigation period, respectively. The best-fit trail (i.e., the optimized parametrization for shrubland) is the trail with minimal $RMSEn$.

The equifinality for the 160 sets of shrub parameters might be an issue, especially in such a non-linear system with many physical processes. We mitigated the equifinality problem by considering two aspects: (1) For the sensitivity analysis method, we used the Morris method - a global sensitivity analysis method. On the one hand, the Morris method samples parameter values from a given interval in a large parameter space (i.e., $16^7$ in our case). This systematic sampling approach ensures a broad exploration of the parameter space, which can help identify the model's sensitivity to different regions. On the other hand, except for ranking the influence of parameters based on elementary effect, the Morris method also quantifies interactions between parameters, which helps understand how parameters jointly affect the model output; (2) For the sensitivity analysis result, the use of objective function $RMSEn$ in requiring RMSE in SWC, LE and GPP can help avoid this problem. More samplings and analysis can be done but is beyond the focus of this study.

9. *Line 218, does it mean Vcmax of shrub and grass is the same (120)? Why?*

**Response 9:** Yes, in this work, we assume the maximum carboxylation rate (Vcmax) is the same for both shrubs and grasses. In light of the reference values from studies involving similar shrub species, Vcmax = 120 for shrub was determined (Wang et al., 2017). For the grassland, the default value for C3 grassland is 80 in SCOPE model and we did not find any reference value for similar species or in similar study area. We assume the same Vcmax for grass as shrub in order to maintain consistency for the species grown in the same study area, regarding their adaptability to the arid region. Besides, the grassland simulation serves as a reference scenario pre-revegetation. The Vcmax variation in the grassland simulation doesn't impact the primary goal: comparing grassland and shrub-grassland scenarios. But indeed, a more representative Vcmax could improve the physical interpretability when comparing two scenarios.

10. *Line 237, the sensors were installed under the grassland, but the simulated soil water content is the average of shrub, grass, and bare soil. So, direct compassion of them may have a large bias.*

**Response 10:** We fully agree on this and will point out the bias in the discussion in a more detailed manner. Since the soil water content is a state variable, which is not reasonable to aggregate/average from shrub, grasses and bare soil simulation in current modelling scheme. Therefore, we ended up with comparing the observed soil moisture with the simulated soil moisture from the grassland simulation.

11. *Line 373, why did not the author attempt to modify the model to simulate evaporation from the bare soil?*

**Response 11:** We did not simulate the evaporation or soil moisture from bare soil individually because STEMMUS-SCOPE considers the soil-root-canopy continuum, and quantifies the amount of energy received and water evaporated based on the leaf area index, gap fraction and leaf inclination. The key idea of this study is to compare the difference in the fluxes between grassland and shrubs-grassland scenario, in order to represent the effects of planting shrubs. Hence, we thought the current modelling scheme adequately represents fluxes in a mixed vegetated area. However, it's worth noting that the STEMMUS model itself can simulate evaporation from bare soil effectively (Zeng et al., 2011), while STEMMUS-SCOPE is more adaptable for vegetated areas (LAI > 0). To address this issue, future model improvements will aim to allow the option of switching on/off the vegetation module as needed.

12. *Line 403. Why cannot the model capture the wet deep soil layer? Is it related to root water uptake? More analysis and simulation should be performed.*

**Response 12:** Sorry if this was unclear in the manuscript. According to the observed dry surface (i.e., low SWC at 10 cm) and a higher observed LE and GPP compared to the simulations, we assumed that the shrubs could switch their root water uptake strategy by either accessing the deep soil water and/or accessing water by lateral roots. If the former assumption is met, our model does not have this option for the plant to switch RWU under specified condition but only simulate the root length growth and RWU by root parameters mentioned in **Response 7.** For the latter assumption of lateral roots, our model is a 1-D vertical model therefore we overlook this process, which is an important survival strategy of shrubs in the study area.

For the RWU simulation, the uncertainties were raised from (1) the setting of the initial soil profile. As shown in the (user-defined) initial conditions (Table S2), the soil moisture under 10 cm was estimated without support from observations during 2016-2019. In the STEMMUS-SCOPE, the initial SWC profile not only determines the pattern of soil water storage but also indicates the pattern of root water uptake in the soil column. That's why we mentioned *"the model might not capture the wet deep soil layer"*; (2) The uncertainties in reconstructed LAI; (3) The uncertainties in estimated Vcmax and root parameters.

However, these are just plausible assumptions because of the lack of observed data from underground. Future analysis will be conducted by investigating the root parameters and modules, to gain more insights and clarifications in the revised manuscript.

**References**

Dan, Y.: Effects of Planted Shrub Encroachment on Evapotranspiration in Desert Steppe ——A Case Study in Yanchi County, Ningxia Hui Autonomous Region (In Chinese), Ningxia University, https://doi.org/10.27257/d.cnki.gnxhc, 2020.

Dan, Y., Du, L., Wang, L., Ma, L., Qiao, C., Wu, H., and Meng, C.: Effects of planted shrub encroachment on evapotranspiration and its components in desert steppe: A case study in Yanchi county, Ningxia Hui Autonomous Region, Shengtai Xuebao/ Acta Ecol. Sin., 40, 5638–5648, https://doi.org/10.5846/STXB201910032066, 2020.

Fensholt, R. and Sandholt, I.: Evaluation of MODIS and NOAA AVHRR vegetation indices with in situ measurements in a semi-arid environment, Int. J. Remote Sens., 26, 2561–2594, https://doi.org/10.1080/01431160500033724, 2005.

Fensholt, R., Sandholt, I., and Rasmussen, M. S.: Evaluation of MODIS LAI, fAPAR and the relation between fAPAR and NDVI in a semi-arid environment using in situ measurements, Remote Sens. Environ., 91, 490–507, https://doi.org/10.1016/j.rse.2004.04.009, 2004.

Jackson, R. B., Mooney, H. A., and Schulze, E. D.: A global budget for fine root biomass, surface area, and nutrient contents, Proc. Natl. Acad. Sci. U. S. A., 94, 7362–7366, https://doi.org/10.1073/PNAS.94.14.7362, 1997.

Kennedy, D., Swenson, S., Oleson, K. W., Lawrence, D. M., Fisher, R., Lola da Costa, A. C., and Gentine, P.: Implementing Plant Hydraulics in the Community Land Model, Version 5, J. Adv. Model. Earth Syst., 11, 485–513, https://doi.org/10.1029/2018MS001500, 2019.

Wang, H., Harrison, S. P., Prentice, I. C., Yang, Y., Bai, F., Togashi, H. F., Wang, M., Zhou, S., and Ni, J.: The China Plant Trait Database, PANGAEA, https://doi.org/10.1594/PANGAEA.871819, 2017.

Wang, Y., Zeng, Y., Yu, L., Yang, P., Van der Tol, C., Yu, Q., Lü, X., Cai, H., and Su, Z.: Integrated modeling of canopy photosynthesis, fluorescence, and the transfer of energy, mass, and momentum in the soil-plant-Atmosphere continuum (STEMMUS-SCOPE v1.0.0), Geosci. Model Dev., 14, 1379–1407, https://doi.org/10.5194/gmd-14-1379-2021, 2021.

Yang, W., Wang, Y., He, C., Tan, X., and Han, Z.: Soil Water Content and Temperature Dynamics under Grassland Degradation: A Multi-Depth Continuous Measurement from the Agricultural Pastoral Ecotone in Northwest China, Sustain. 2019, Vol. 11, Page 4188, 11, 4188, https://doi.org/10.3390/SU11154188, 2019.

Zeng, Y., Su, Z., Wan, L., and Wen, J.: Numerical analysis of air-water-heat flow in unsaturated soil: Is it necessary to consider airflow in land surface models?, J. Geophys. Res. Atmos., 116, D20107, https://doi.org/10.1029/2011JD015835, 2011.

---

## Author Response (AR1)

**Response to associate editor decision**

Dear Dr. Akihiko Ito,

We sincerely thank you for your thoughtful review of our manuscript and responses to the referees. It is with great pleasure to have the opportunity to submit a revised manuscript to Biogeosciences. Your commitment of time in providing valuable feedback on our work is greatly appreciated. Enclosed below is our response addressing your comment.

**Comment:** *Concerning Comment 5 from Referee #2, i.e., spin-up of the model. You wrote that the model adopted observational data as the initial condition and did not perform a spin-up. I agree that this applies to soil moisture but have a concern about carbon stock and LAI, because most models require a long spin-up to obtain stable vegetation and soil carbon state. Please clarify this point.*

**Response:** While it holds true that spin-up procedures are beneficial for establishing stable vegetation and soil carbon states, it is important to note that the STEMMUS-SCOPE model does not involve the simulation of carbon stocks, including LAI. Just as with soil moisture, we utilized observed LAI values as the model input, which served well as a stable vegetation state. Our model primarily focuses on simulating carbon assimilation within the existing framework. Consequently, we do not consider the spin-up (to attain a stable LAI and soil carbon state), primarily due to the absence of carbon allocation in our model. We are aware of this issue and is addressing it in our upcoming study by coupling the STEMMUS-SCOPE model with the WOFOST crop growth model. This coupling will allow us to simulate LAI and plant growth processes effectively. The evaluation of the coupled STEMMUS-SCOPE-WOFOST model, performed with various plant functional types, showed successful simulations of vegetative dynamics, including LAI and plant height. We intend to submit the study of the STEMMUS-SCOPE-WOFOST model to a journal soon.

Thanks a lot for your attention and consideration! We look forward to the insightful feedback and discussions on our revised manuscript.

Yours sincerely,

Enting Tang, on behalf of Zhongbo Su, Yijian Zeng, Lingtong Du

**Point-by-point response to the reviews including a list of all relevant changes**

Only the comments with relevant changes in the revised manuscript are listed below. Elaboration on other comments from Referee#1 and Referee#2 have been addressed in the open discussion platform https://bg.copernicus.org/preprints/bg-2023-70/#discussion.

**Comments from Referee#1**

*1. In the introduction section, the scientific question is not clear. Generally, the authors thought root water uptake is a critical process in the modeling, and the dynamic root length density for estimating root water uptake is necessary. However, no contents about the root water uptake were presented in this study. What is the impact of dynamic root length density? What is the performance of root water uptake simulation? This is the major limitation of this study.*

*7. Why were root-related parameters not identified as influential parameters? This is the main focus of your study.*

**Response:** Thank you for the insightful suggestions, and the following points were addressed in the revised manuscript.

(1) Lines 59-65 were added to elaborate the research gap further, thus drawing out the need to explore the soil water and root water uptake in different layers by applying the STEMMUS-SCOPE model.

(2) Lines 269-274 and Lines 434-435 were added to analyze root water uptake simulation.

(3) The dynamic root length density defines the relative amount of root water uptake in each soil layer. Therefore, we directly compared the difference in RWU, which is simulated based on maximum rooting depth, fitted extinction coefficient, initial root density and root biomass. The impacts of maximum rooting depth, fitted extinction coefficient and initial root density were evaluated in the sensitivity analysis (Lines 202-207). We found that the root-related parameters are more influential in the simulations of soil water content and ground heat flux and have a relatively strong interactive effects in all the simulated fluxes (Fig. S5 in Supplement).

*8. How did the authors optimize the parameters (best-fit trail in Line 196)?*

**Response:** Incorporating three root parameters, we updated the sensitivity analysis results where 220 sets of parameters for shrubland were generated. The determination of the optimal trail is described in Lines 181-186 in the revision. The updated parameters were updated in Lines 205-206. The optimal trail was adapted for grassland scenario with adjustments to root parameters as detailed in Table S1.

*10. The sensors were installed under the grassland, but the simulated soil water content is the average of shrub, grass, and bare soil. So, direct compassion of them may have a large bias.*

**Response:** A more detailed discussion on this bias was updated in Lines 351-356.

*12. Why cannot the model capture the wet deep soil layer? Is it related to root water uptake? More analysis and simulation should be performed.*

**Response:** It was an unclear statement in the first manuscript. From Lines 377 to 388 in the revised version, we rephrased the analysis of the uncertainties in simulating LE and GPP. In conclusion, the underestimations might be caused by the uncertainties in the approximated LAI, $V_{cmax}$ and initial setting of the SWC profile. The root parameters are not the main influential parameters to LE and GPP simulations, while they are found to be more influential to the simulations of soil moisture and ground heat flux. Because the root growth/distribution directly interacts with the soil water content/distribution in STEMMUS-SCOPE calculation.

**Comments from Referee#2**

The title was changed into *"Understanding the Effects of Revegetated Shrubs on fluxes of Energy, Water and Gross Primary productivity in a Desert Steppe Ecosystem Using STEMMUS-SCOPE Model"*.

*1. Equation (1), what does C_shrub mean?*

**Response:** We have added an explanation of this variable in Line 161.

*6. Figure 4c, there are a few points aligning as a vertical line. Does it indicate there are some problems with the data or the model?*

**Response:** The cause of the vertical line has been explained in the author's response in the open discussion platform. Accordingly, we summarized this uncertainty in Lines 356-360.

*8. Figure 6, could you also show the variability?*

**Response:** The variability was added in the updated plot (Figure 6, Line 275).

*9. Section 3.3.3 on GPP, I don't think the interpretation of midday depression is accurate. It may involve factors more than radiation. Perhaps you can show the diurnal pattern of leaf temperature, and temperature dependence of carboxylation as well.*

**Response:** The diurnal pattern of leaf temperature was added in Figure S10 (c) in the revised Supplement. And the interpretation of midday depression was improved as written in Line 418-423.

*10. Discussion 4.2.1, paragraph 2, why don't you use the Bowen ratio as an indicator, which is likely more informative than the ratio of LE/Rn here.*

**Response:** The Bowen ratio was calculated and updated in Line 398 and indeed is a better indicator to tell the dominance between sensible heat flux and latent heat flux.

---

## Author Response (AR2)

**Response to associate editor decision – Minor revision**

Dear Dr. Akihiko Ito,

We are deeply grateful for your thorough review and insightful comments on the revised manuscript. Your comments are valuable in enhancing the quality of the manuscript. Enclosed below is our response addressing your comment.

**Comment:** *I studied the discussion and the manuscript and confirmed that most technical points were revised adequately. However, I have still a small concern about your response to the specific comment #1 by referee #1 saying that "the scientific question is not clear". Your response focused on technical points about root water uptake and did clarify what is the scientific question. So, I recommend giving a clear scientific question of this study by revising sentences at around Line 63 of the manuscript; for example, "We need a mechanistic understanding on the ecohydrological process, especially soil water movement in different layers and its connection with root water uptake (RWU) and photosynthesis. Therefore, we applied …". This amendment would explain the scientific significance of this study which the referee #1 felt some weakness.*

**Response:** We made the amendment at Line 63 – Line 67 according to your suggestion. Additionally, we concluded three key research questions at Line 73 – Line 77 to clarify the scientific significance of this study.

Thanks a lot for your consideration! We look forward to the insightful feedback and discussions on our revised manuscript.

Yours sincerely,

Enting Tang, on behalf of Zhongbo Su, Yijian Zeng, Lingtong Du

---

## Author Response (AR3)

**Response to associate editor decision: Publish as is**

Dear Dr. Akihiko Ito,

Thank you for your positive feedback and for recommending our manuscript for publication. We are pleased to hear that the revisions we made have effectively enhanced the manuscript's clarity. Your constructive comments have been invaluable in this process. Enclosed below are the relevant changes made in the manuscript for production.

**Remarks from the preceding review file validation:** *Regarding your figure 1: with the next revision, please update the copyright statement as follows: (map © OpenStreetMap contributors YEAR. Distributed under the Open Data Commons Open Database License (ODbL) v1.0.)*

**Response:** We adjusted the copyright icon "©" in the map itself as required by the Editorial Support Team in June. In this revision, we have added the copyright statement to the caption of Figure 1 as follows (see Line 101 in the manuscript):

Figure 1. Overview of the study site. The red area is Yanchi County in China (Map source: OpenStreetMap contributors 2021. Distributed under the Open Data Commons Open Database License (ODbL) v1.0). The white star denotes the field station located in Yangzhaizi village in Yanchi County, Ningxia Province.

Besides, we added the number "(2)" to indicate the equation at Line 189. Throughout the text, some typos and the use of hyphens instead of en dashes has been corrected.

We appreciate the time and effort from you, referees and editorial officers to review our work, and we are excited about the opportunity to contribute to the field through this publication. Thank you once again for your support and recommendation.

Yours sincerely,

Enting Tang, on behalf of Zhongbo Su, Yijian Zeng, Lingtong Du